# ProbGAN: Towards Probabilistic GAN with Theoretical Guarantees

**Hao He, Hao Wang, Guang-He Lee, Yonglong Tian**
Computer Science and Artificial Intelligence Laboratory
Massachusetts Institute of Technology
{haohe,hwang87,guanghe,yonglong}@mit.edu

## Abstract

Probabilistic modelling is a principled framework to perform model aggregation, which has been a primary mechanism to combat mode collapse in the context of Generative Adversarial Networks (GAN). In this paper, we propose a novel probabilistic framework for GANs, *ProbGAN*, which iteratively learns a distribution over generators with a carefully crafted prior. Learning is efficiently triggered by a tailored stochastic gradient Hamiltonian Monte Carlo with a novel gradient approximation to perform Bayesian inference. Our theoretical analysis further reveals that our treatment is the first probabilistic framework that yields an equilibrium where generator distributions are faithful to the data distribution. Empirical evidence on synthetic high-dimensional multi-modal data and image databases (CIFAR-10, STL-10, and ImageNet) demonstrates the superiority of our method over both start-of-the-art multi-generator GANs and other probabilistic treatment for GANs.

## 1 Introduction

Generative Adversarial Networks (GAN) (Goodfellow et al., 2014) is notoriously hard to train and suffers from *mode collapse*. There has been a series of works attempting to address these issues. One noticeable thread focuses on objective design, which improves the original Jensen-Shannon divergence with more stable pseudo-metrics such as $f$-divergence (Nowozin et al., 2016), $\chi^2$-divergence (Mao et al., 2017), and Wasserstein distance (Arjovsky et al., 2017). However, such treatment is inherently limited when a single generator does not include enough model capacity to capture the granularity in data distribution in practice. Clearly, such a generator can hardly produce accurate samples regardless of the choice of objectives.

An alternative remedy is to learn multiple generators instead of a single one. This type of methods (Hoang et al., 2018; Tolstikhin et al., 2017; Wang et al., 2016b) is motivated by a straightforward intuition that multiple generators can better model multi-modal distributions since each generator only needs to capture a subset of the modes. To entail model aggregation, probabilistic modelling is a natural and principled framework to articulate the aggregation process.

Recently, Saatci & Wilson (2017) propose Bayesian GAN, a probabilistic framework for GAN under Bayesian inference. It shows that modelling the distribution of generator helps alleviate mode collapse and motivates the interpretability of the learned generators. This probabilistic framework is built upon Bayesian models for generator and discriminator, whose maximum likelihood estimation can be realized as a metaphor of typical GAN objectives.

While empirical study on semi-supervised image classification tasks shows the effectiveness of Bayesian GAN, a critical theoretical question on this framework remains unanswered: Does it really converge to the generator distribution that produces the real data distribution? Indeed, our theoretical analysis and experimental results on a simple toy dataset reveal that the current Bayesian GAN falls short of convergence guarantee.

With this observation, we follow the prior work to exploit probabilistic modelling as a principled way to realize model aggregation, but approach this problem from a theoretical perspective. We analyze the developed treatment, including the choice of priors, approximate inference, as well as its

Table 1: Common GAN objective functions.

|  | $\phi_1(D)$ | $\phi_2(D)$ | $\phi_3(D)$ | MIN-MAX STYLE |
|---|---|---|---|---|
| GAN (MIN-MAX) | $\log(D)$ | $\log(1-D)$ | $-\log(1-D)$ | YES |
| GAN (NON-SATURATING) | $\log(D)$ | $\log(1-D)$ | $\log(D)$ | NO |
| WASSERSTEIN GAN | $D$ | $-D$ | $D$ | YES |
| LEAST-SQUARES GAN | $(D-1)^2$ | $D^2$ | $(D-1)^2$ | NO |

convergence property, and simultaneously propose a new probabilistic framework with the desirable convergence guarantee and consequently superior empirical performance.

Our main contributions are:

- We theoretically establish, to our best knowledge, the first probabilistic treatment of GANs such that any generator distribution faithful to the data distribution is an equilibrium.

- We prove the previous Bayesian method (Saatci & Wilson, 2017) for any minimax GAN objective induces incompatibility of its defined conditional distributions.

- We propose two special Monte Carlo inference algorithms for our probabilistic model which efficiently approximate the gradient of a non-differentiable criterion.

- Empirical studies on synthetic high-dimensional multi-modal data and benchmark image datasets, CIFAR-10, STL-10, and ImageNet, demonstrate the superiority of the proposed framework over the state-of-the-art GAN methods.

## 2 RELATED WORK

**Generative Adversarial Networks** is a powerful class of methods to learn a generative model for any complex target data distribution. There is a game between a generator and a discriminator. Both of them adapt their strategies to maximize their own objective function involving the other:

$$
\begin{aligned}
\max_{\theta_d} \mathcal{J}_d(\theta_d; \theta_g) &= \mathbb{E}_{\mathbf{x} \sim p_{data}}[\phi_1(D(\mathbf{x}; \theta_d)])] + \mathbb{E}_{\mathbf{x} \sim p_{gen}(\cdot; \theta_g)}[\phi_2(D(\mathbf{x}; \theta_d))], \\
\max_{\theta_g} \mathcal{J}_g(\theta_g; \theta_d) &= \mathbb{E}_{\mathbf{x} \sim p_{gen}(\cdot; \theta_g)}[\phi_3(D(\mathbf{x}; \theta_d))].
\end{aligned}
\tag{1}
$$

Eqn. 1 gives a general mathematical form where $p_{data}$ is real data distribution and $p_{gen}(\cdot; \theta_g)$ are generated data distribution with generator parameter $\theta_g$. The objective functions $\phi_1, \phi_2, \phi_3$ (termed as *GAN objective* in this paper) are elaborately chosen such that at the equilibrium, the generator generates the target data distribution. Table 1 summarizes several widely used GAN objectives, including the original min-max version, non-saturating version of original GAN (Goodfellow (2016)), LSGAN (Mao et al. (2017)), and WGAN (Arjovsky et al. (2017)). As reported in Table 1, some GAN objectives, satisfying $\phi_3(\cdot) = -\phi_2(\cdot)$, actually represent a min-max game, i.e. $\min_{\theta_g} \max_{\theta_g} \mathcal{J}_d(\theta_d; \theta_g)$.

**Training GAN with multiple generators** is considered in several recent works to mitigate the mode collapse problem. In the spirit of boosting algorithm, Wang et al. (2016b) propose to progressively train new generator using a subset of training data that are not well captured by previous generators, while Tolstikhin et al. (2017) further propose a more robust mechanism to reweight samples in the training set for a new generator. From the perspective of game theory, MIX-GAN (Arora et al., 2017) extends the game between a single generator and discriminator to the multiple-player setting. Other works resort to third-party classifiers to help multiple generators and discriminators achieve better equilibrium, such as MAD-GAN (Ghosh et al., 2017) and the recent state-of-art method, MGAN (Hoang et al., 2018).

**Bayesian GAN** proposed by Saatci & Wilson (2017) adopts a different approach which models generator and discriminator distributions by defining the conditional posteriors (Eqn. 8). The likelihood model is specially designed such that maximizing it exactly corresponds to optimizing GAN objectives. The authors argue that compare to point mass ML estimation, learning the generator distribution which is multi-modal itself offers better ability to fit a multi-modal data distribution.

To facilitate discussion, we categorize GAN frameworks into the following taxonomy: *optimization-based methods* and *probabilistic methods*. Optimization-based methods set up an explicit mini-max game between the generator and discriminator, where an ideal equilibrium typically characterize a generator faithful to data distribution. In probabilistic methods, generators and discriminators

evolve as particles of underlying distributions, where an equilibrium is searched from a stochastic exploration in the distribution space (of the generators and discriminators).

# 3 METHODOLOGY

We first summarize the notations. Second, we elaborate *ProbGAN*, our probabilistic modelling for GAN, and introduce its Bayesian interpretation by developing constituent prior and likelihood formulations. Finally, we develop inference algorithms for ProbGAN. A detailed discussion of the motivation of our modelling and the comparison with Bayesian GAN is included in Section 4.

## 3.1 NOTATIONS

$p_{data}(\mathbf{x})$ over a sample space $\mathcal{X}$ is the target data distribution we want to learn. Our generator and discriminator are parameterized by $\theta_g \in \Theta_g$ and $\theta_d \in \Theta_d$. A generator with parameter $\theta_g$ defines a mapping from a random noise vector $\mathbf{z} \sim p_z$ to a random vector $G(\mathbf{z}; \theta_g)$. The induced probability density of $G(\mathbf{z}; \theta_g)$ is denoted as $p_{gen}(\mathbf{x}; \theta_g)$. A discriminator is a function that maps data to a real-valued score, i.e. $D(\mathbf{x}; \theta_d) : \mathcal{X} \to [0, 1]$ (or $\mathcal{X} \to \mathbb{R}$ in some settings). Further, we use $q_g(\theta_g) \in \mathcal{P}^{\Theta_g}$, $q_d(\theta_d) \in \mathcal{P}^{\Theta_d}$ to denote the distribution over generators and discriminators respectively.

The total data distribution generated by generator following the density $q_g(\theta_g)$ is naturally a mixture of data distribution given by every single generator, $p_{model}(\mathbf{x}; q_g) = \mathbb{E}_{\theta_g \sim q_g(\theta_g)}[p_{gen}(\mathbf{x}; \theta_g)]$. Our goal is to find a generator distribution $q_g^*(\theta_g)$ such that the total generated data distribution matches our target, i.e. $p_{model}(\mathbf{x}; q_g^*) \simeq p_{data}(\mathbf{x})$.

$\mathcal{J}_g(\theta_g; \theta_d)$ and $\mathcal{J}_d(\theta_d; \theta_g)$ denote objective functions of generator and discriminator as introduced in Eqn. 1. The common choices[1] are listed in Table 1. With a slight abuse of the notation, we extend the notation $\mathcal{J}_g(\theta_g; D^*)$ by replacing $D(\mathbf{x}; \theta_d)$ in equation 1 with any score function $D^*$. Then $\mathcal{J}_g(\theta_g; \theta_d)$ can be viewed as an abbreviation of $\mathcal{J}_g(\theta_g; D(\cdot; \theta_d))$. Likewise, $\mathcal{J}_d(\theta_d; p_{gen}(\cdot))$ represents discriminator objective given a virtual generator that generates data with density $p_{gen}(\cdot)$.

## 3.2 PROBGAN

Like Bayesian GAN, ProbGAN learns distributions of the generator and the discriminator. During training, the target data distribution is given (by samples) and treated as an fixed environment. While for the generator/discriminator, they observe each other and adapt their own parameters based on the observation. To facilitate the comparison to Bayesian GAN, we state ProbGAN in a Bayesian formulation. Every generator/discriminator distribution update can be viewed as the following posterior inference process.

**Posterior.** ProbGAN updates generator/discriminator distributions based on their distributions in previous time step and the target data distribution as shown in Eqn. 2. We we denote $q_g^{(t)}$ and $q_d^{(t)}$ as distributions at time step $t$.

$$q_g^{(t+1)}(\theta_g) \propto \exp\{\mathcal{J}_g(\theta_g; D^{(t)})\} \cdot q_g^{(t)}(\theta_g), \quad q_d^{(t+1)}(\theta_d) \propto \exp\{\mathcal{J}_d(\theta_d; p_{model}^{(t)})\}. \tag{2}$$

To understand it is a posterior modelling, we further interpret the terms in Eqn. 2 as likelihood term and prior term seperately.

**Likelihood.** We call the exponential terms in Eqn. 2 as likelihood terms.

$$p(\theta_g) \propto \exp\{\mathcal{J}_g(\theta_g; D^{(t)})\}, \quad p(\theta_d) \propto \exp\{\mathcal{J}_d(\theta_d; p_{model}^{(t)})\}. \tag{3}$$

where $p_{model}^{(t)}(\mathbf{x}) = \mathbb{E}_{\theta_g \sim q_g^{(t)}}[p_{gen}(\mathbf{x}; \theta_g)]$ is the mixed data distribution under the current generator distribution $q_g^{(t)}$ and $D^{(t)}(\cdot) = \mathbb{E}_{\theta_d \sim q_d^{(t)}}[D(\cdot; \theta_d)]$ is the averaged discriminating score function under current discriminator distribution $q_d^{(t)}$.

---

[1] The concepts of minimax version and non-saturating version of vanilla GAN are first introduced in Goodfellow (2016).

These likelihoods indicate a preference for generators and discriminators, given current distributions of generator and discriminator. More specifically, likelihoods in Eqn. 3 encode the information that distributionally reflect the objective of generators $\mathcal{J}_g$ and discriminators $\mathcal{J}_d$. Such quantities evaluate the fitness between the generator and the discriminator.

We emphasize, although sharing the same spirit of reflecting the GAN objectives in likelihood, there is a crucial difference between our likelihood model and that of Bayesian GAN. We will revisit it in the later theory section.

**Prior.** Unlike Bayesian GAN using normal distributions for both generator and discriminator, ProbGAN has less standard priors. As Eqn. 2 suggests, we set different priors for the two players. For the generator, we use the generator distribution in the previous time step as a prior. The intuition is following. When the generated data distribution is increasingly close to the real data distribution, there will be less information for discriminator to distinguish between them; consequently, the discriminator tends to assign equal scores to all data samples, resulting in equal likelihoods for all generators. At that stage, a good strategy is to keep the generator distribution the same as the previous time step, since it already generates the desired data distribution perfectly. Hence, we use the generator distribution in the previous time step as a prior for the next Such dynamically evolving prior for generator turns out to be crucial. In Section 4.2, we show the Bayesian GAN suffers from bad convergence due to its fixed and weakly informative prior. In contrast, we set a uniform improper prior on the discriminator to pursuit unrestricted adaptability to evolving generators.

### 3.3 INFERENCE ALGORITHM

So far we have introduced our ProbGAN model. In this section we develop novel inference algorithms to compute the posterior. Similar to most advanced Bayesian methods, exact calculation of the posterior is *intractable*. Following the strategy in Saatci & Wilson (2017), we adopt Stochastic Gradient Hamiltonian Monte Carlo to generate samples from the posterior. In each iteration, $M$ samples $\{\theta_{g,m}^{(t)}\}_{m=1}^{M}$ are used to approximate the generator distribution $q_g^{(t)}$.

$$\nabla_{\theta_d} \log q_d^{(t+1)}(\theta_d) = \nabla_{\theta_d} \mathcal{J}_d(\theta_d; p_{model}^{(t)}) \approx \frac{1}{M_g} \sum_{m=1}^{M_g} \nabla_{\theta_d} \mathcal{J}_d(\theta_d; p_{gen}(\cdot; \theta_{g,m}^{(t)})), \quad (4)$$

$$\nabla_{\theta_g} \log q_g^{(t+1)}(\theta_g) = \nabla_{\theta_g}(\mathcal{J}_g(\theta_g; D^{(t)}) + \log q_g^{(t)}(\theta_g)) \approx \nabla_{\theta_g}(\mathcal{J}_g(\theta_g; \frac{1}{M_d} \sum_m D(\cdot; \theta_{d,m}^{(t)})) + \log q_g^{(t)}(\theta_g)). \quad (5)$$

Empowered by the adapted SGHMC (Algorithm 1 in the appendix), we are able to sample from $q_g^{(t+1)}$ and $q_d^{(t+1)}$ based on gradients in Eqn. 4 and Eqn. 5. The gradients come from two sides: the GAN objective $\mathcal{J}_g$, $\mathcal{J}_d$ and the prior $q_g^{(t)}$. Getting GAN objective's gradient is easy while computing the prior's gradient, $\nabla_{\theta_g} \log q_g^{(t)}(\theta_g)$, is actually non-trivial since we have no exact analytic form of $q_g^{(t)}(\theta_g)$. To address this challenge, we propose the following two methods to approximate $\nabla_{\theta_g} \log q_g^{(t)}(\theta_g)$, leading to two practical inference algorithms.

**Gaussian Mixture Approximation (GMA).** Although the analytic form of the distribution $q_g^{(t)}(\theta_g)$ is unknown, we have $M_g$ Monte Carlo samples $\{\theta_{g,m}^{(t)}\}_{m=1}^{M_g}$ which enables us to directly approximate the distribution as a Mixture of Gaussian in the left side of Eqn. 6, where $\sigma$ is a hyper-parameter and C is the normalization constant. Then we derive the prior gradient approximation as shown in the right side of Eqn. 6.

$$q_g^{(t+1)}(\theta_g) \approx C \exp\{\sum_{m=1}^{M_g} \frac{\|\theta_g - \theta_{g,m}^{(t)}\|_2^2}{2\sigma^2}\} \Rightarrow \nabla_{\theta_g} \log q_g^{(t+1)}(\theta_g) \approx \sum_{m=1}^{M_g} \frac{1}{\sigma^2}(\theta_g - \theta_{g,m}^{(t)}). \quad (6)$$

**Partial Summation Approximation (PSA).** From Eqn. 5, actually we can make an interesting observation that the prior gradient can be recursively unfolded as a summation over all historical GAN objective gradients, shown as:

$$\nabla_{\theta_g} \log q_g^{(t+1)}(\theta_g) = \nabla_{\theta_g} \mathcal{J}_g(\theta_g; D^{(t)}) + \nabla_{\theta_g} \log q_g^{(t)}(\theta_g) = \sum_{i=0}^{t} \nabla_{\theta_g} \mathcal{J}_g(\theta_g; D^{(i)}). \quad (7)$$

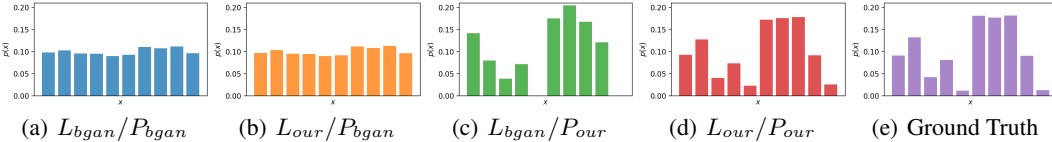

(a) $L_{bgan}/P_{bgan}$    (b) $L_{our}/P_{bgan}$    (c) $L_{bgan}/P_{our}$    (d) $L_{our}/P_{our}$    (e) Ground Truth

Figure 1: An example of data distributions produced by converged models in the toy experiment on categorical distribution. We examined four possible combinations of likelihoods and priors. $L_{our}$, $L_{bgan}$ stand for the likelihoods of our ProbGAN model and BGAN model. $P_{our}$ and $P_{bgan}$ stand for the priors. Only our model (Figure 1(d)) learns the target data distribution (Figure 1(e)).

Therefore if we store all historical discriminator samples $\{\theta_{d,m}^{(i)}\}_{i=1,m=1}^{t,M_d}$, the prior gradient can be computed accurately via simple summation. Practically, computing gradients with all discriminator samples costs huge amount of storage and computational time, which is unaffordable. Hence we propose to maintain a subset of discriminators by subsampling the whole sequence of discriminators.

## 4 THEORY

In this section, we first present the good convergence property of ProbGAN. Second, we theoretically analyze the distribution evolution of Bayesian GAN (which will be referred to as BGAN in the rest of the paper) and compare BGAN with ProbGAN. All proofs are included in the appendix (Section A).

### 4.1 CONVERGENCE PROPERTY OF PROBGAN

We say a generator distribution is ideal if the generator following this distribution produces the target data distribution. Theorem 1 shows that any ideal generator distribution is an equilibrium of the dynamics defined in Eqn. 2. Although its mathematical proof involves more elaboration, the idea behind is quite simple. When the generator distribution is ideal, the discriminator is not able to distinguish the synthetic data from real. Thus the averaged discriminator function will degenerate to a constant function. Afterwards, the generator distribution will remain unchanged since the discriminator essentially puts no preference over generators. Here, we note that the discriminator is only involved in the likelihood. The prior still needs to be carefully designed so that the model can converge to an equilibrium where the generator is ideal. Simply choosing a weakly informative prior as Bayesian GAN did will not give Theorem 1.

**Theorem 1.** *Assume the GAN objective and the discriminator space are symmetry. For any* ideal *generator distribution $q_g^*(\theta_g)$ satisfying $p_{model}^* \triangleq \mathbb{E}_{\theta_g \sim q_g^*}[p_{gen}(\cdot; \theta_g)] = p_{data}$, there exists a discriminator distribution $q_d^*$ such that $D^*(\cdot) \triangleq \mathbb{E}_{\theta_d \sim q_d^*} D(\cdot; \theta_d) \equiv \frac{c}{2}$. Moreover, $q_g^*$ and $q_d^*$ is an equilibrium of the dynamic defined in Eqn. 2.*

### 4.2 PROBGAN V.S. BAYESIAN GAN

This section presents analyses of the BGAN algorithm, where we find a theoretical issue in its convergence and highlight the importance of our renovation of the prior and likelihood.

Corollary 1 states our derivation of posterior modelling in BGAN, where $p_g(\theta_g; \alpha_g), p_d(\theta_d; \alpha_d)$ are the predefined priors. In practice, BGAN use a fixed Gaussian prior.

**Corollary 1.** *The Bayesian GAN algorithm actually performs distribution dynamics in Eqn. 8.*

$$q_g^{(t+1)} \propto \exp\{\mathbb{E}_{\theta_d \sim q_d^{(t)}} \mathcal{J}_g(\theta_g; \theta_d)\} p_g(\theta_g; \alpha_g), q_d^{(t+1)} \propto \exp\{\mathbb{E}_{\theta_g \sim q_g^{(t)}} \mathcal{J}_d(\theta_d; \theta_g)\} p_d(\theta_d; \alpha_d) \quad (8)$$

**Difference in Likelihood.** The subtle adjustment of our likelihood term lies in the order of taking expectation. As shown in Eqn. 3, our choice of likelihood yields a concrete physical meaning. Our discriminator likelihood explicitly evaluates the discriminator ability of distinguishing real data distribution and total data distribution generated by all generators. Hence, our approach matches the target data distribution with mixed data distribution produced by generators, while BGAN may not.

**Difference in Prior.** The choice of prior also plays an important role in convergence. A weakly/non-informative prior, as adopted in BGAN, prevents the generators from convergence even if they already produce the data distribution faithfully. The phenomenon arises from the fact that the information provided by discriminator vanishes when the generators are ideal and the resulting generator posterior

will degenerate to the prior. To remedy this issue, our solution is to take generator distribution at previous time step as the prior. Then whenever the discriminator degenerates to a constant, the generator distribution will stay unchanged because the prior is itself.

**An Analytical Case Study.** We demonstrate the superior convergence property of our model on a categorical distribution, where analytic posterior computation of various choices of likelihood and prior are feasible; we compute the exact equilibrium of GAN models under all the four combinations of the priors and likelihoods(ProbGAN's choices v.s. BGAN's choices). The experiment details are in the appendix (section D). Figure 1 is an example of the data distributions generated by each model after it converges. Among all the combinations, our model is *the only formulation that yields proper convergence*, which validates our theoretical analysis.

**Compatibility Issue**. We further show BGAN's choice of likelihood and prior leads to theoretical issues. Specifically, BGAN is not suitable for any minimax-style GAN objective due to the incompatibility of its conditional posteriors. This problem may limit the usage of BGAN since many widely used GAN objective is in min-max fashion, such as the original GAN and WGAN.

Consider a simple case where we use only one Monte Carlo sample for the distributions $q_g^{(t)}$ and $q_d^{(t)}$. Then the distribution evolution in Eqn. 8 will degenerate to a Gibbs sampling process.

$$
\begin{aligned}
\theta_g^{(t+1)} &\sim q_g^{(t+1)}(\theta_g) \propto \exp\{\mathcal{J}_g(\theta_g; \theta_d = \theta_d^{(t)})\} p(\theta_g | \alpha_g) \\
\theta_d^{(t+1)} &\sim q_d^{(t+1)}(\theta_d) \propto \exp\{\mathcal{J}_d(\theta_d; \theta_g = \theta_g^{(t)})\} p(\theta_d | \alpha_d)
\end{aligned}
\tag{9}
$$

Thus $\theta_g^{(t)}$ and $\theta_d^{(t)}$ are implicitly sampled from a joint distribution of $\theta_d$ and $\theta_g$ defined by the conditionals $p(\theta_g | \theta_d) = \exp\{\mathcal{J}_g(\theta_g; \theta_d)\} p(\theta_g | \alpha_g)$ and $p(\theta_d | \theta_g) = \exp\{\mathcal{J}_d(\theta_d; \theta_g)\} p(\theta_d | \alpha_d)$. However, our theoretical analysis shows that such a presumed joint distribution does not exist when $\mathcal{J}_d(\theta_d; \theta_g) = -\mathcal{J}_g(\theta_g; \theta_d)$. Specifically, Lemma 1 shows the existence of a joint distribution satisfying the conditionals in Eqn. 9 requires the GAN objective to be decomposable, i.e. $\exists \phi_g, \phi_d$, s.t. $\mathcal{J}_d(\theta_d; \theta_g) = \phi_g(\theta_g) + \phi_d(\theta_d)$. Apparently, no valid GAN objective is decomposable. Therefore, conditionals in Eqn. 9 are actually incompatible. Sampling with incompatible conditional distribution is problematic and leads to unpredictable behavior (Arnold & Press, 1989).

**Lemma 1.** *Consider a joint distribution $p(x, y)$ of variable $X$ and $Y$. Its conditional distributions can be represented in the forms of $p(x|y) \propto \exp\{L(x, y)\} q_x(x)$ and $p(y|x) \propto \exp\{-L(x, y)\} q_y(y)$ only if $X$ and $Y$ are independent, i.e., $p(x, y) = p(x)p(y)$ and $L(x, y)$ is decomposable, i.e., $\exists L_x$ and $L_y, L(x, y) = L_x(x) + L_y(y)$.*

## 5 EXPERIMENTS

In this section, we evaluate our model with two inference algorithms proposed in Section 3.3 (denoted as *ProbGAN-GMA* and *ProbGAN-PSA*). We compare with three baselines: 1) *GAN (or DCGAN[2])*: naively trained multiple generators in the vanilla GAN framework; 2) *MGAN*: Mixture GAN (Hoang et al., 2018) which is the start-of-art method to train GAN with multiple generators; 3) *BGAN*: Bayesian GAN (Saatci & Wilson, 2017).

For each model, we conduct thorough experiments with the four different GAN objectives introduced in Table 1, which are referred to as GAN-MM, GAN-NS, WGAN and LSGAN here. For a fair comparison, each model has the same number of generators with the same architecture. Discriminator architectures are also the same except for that of MGAN which has an additional branch of the classifier. To facilitate reproducibility, we report implementation and experiment details in Section B of the appendix.

### 5.1 HIGH-DIMENSIONAL MULTI-MODAL SYNTHETIC DATASET

**Dataset.** Consider learning a data distribution in a high dimensional space $\mathcal{X} = \mathbb{R}^D$, which is a uniform mixture of $n$ modes. Each mode lies on a $d$-dimensional sub-space of $\mathcal{X}$. We call this $d$-dimensional sub-space as mode-space of the $i$-th mode. Specifically, the data of the $i$-th mode is generated by the following process,

---

[2]We use original GAN for synthetic dataset and DCGAN for image generation task

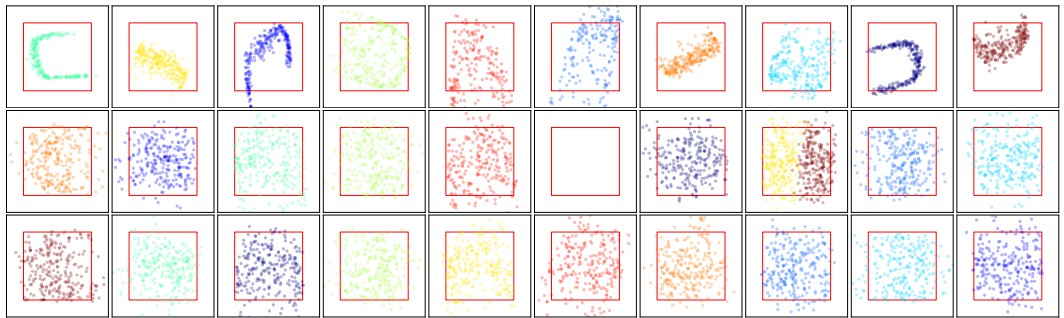

Figure 2: Visualization of projected hit sets. Rows from top to bottom correspond to MGAN, BGAN, and ProbGAN-PSA trained with GAN-MM objective. See Figure 6 (in appendix) for results of all models. In one row, projected hit sets for each mode are plotted in different panels, where the red boxes indicate real data regions $\mathcal{U}[-1, 1]^2$. Different colors indicate samples from different generators.

$$\mathbf{z} \sim \mathcal{U}[-1, 1]^d, \quad \mathbf{x} = \mathbf{A}_i(\mathbf{z} + \mathbf{b}_i), \quad \mathbf{A}_i \sim \mathcal{N}(0, \sigma_A^2 I_{D \times d}), \quad \mathbf{b}_i \sim \mathcal{N}(\mathbf{0}, \sigma_b^2 I_d) \qquad (10)$$

In our experiment, $n$, $D$, and $d$ are set to 10, 100, and 2. Hyper-parameters for $\mathbf{A}$ and $\mathbf{b}$ are set to be $\sigma_A = \sigma_b = 5$. Each model train ten generators.

**Metric.** We define *projection distance* $\epsilon_p$ for generated data sample $\mathbf{x}$ as the minimum of Euclidean distance from $\mathbf{x}$ to any of mode-spaces i.e. $\epsilon_p(\mathbf{x}) = \min_i \epsilon_i(\mathbf{x}) \triangleq \|\mathbf{x} - \mathbf{A}_i(\mathbf{A}_i^T \mathbf{A}_i)^{-1} \mathbf{A}_i^T \mathbf{x}\|_2$. Then we set a threshold [3] $\eta$ to test the belonging of $\mathbf{x}$, i.e. the data samples whose Euclidean distance to the mode-space is below $\eta$ are considered as belonging to that mode. The trained models are evaluated by the samples $\{\mathbf{x}_k\}_{k=1}^K \sim p_{model}$ it generates. We define *hit set* $\mathcal{H}_i \triangleq \{\mathbf{x}_k | \epsilon_i(\mathbf{x}_k) < \eta\}$ to indicate the samples belong to each mode. We further define *projected hit set*, $\mathcal{PH}_i \triangleq \{(\mathbf{A}_i^T \mathbf{A}_i)^{-1} \mathbf{A}_i^T \mathbf{x} - \mathbf{b}_i | \mathbf{x} \in \mathcal{H}_i\}$ by projecting data in each hit set back to the canonical low dimensional space.

Now we introduce three evaluation metrics: *hit ratio*, *hit distance*, and *cover error*. *Hit ratio* $\mathcal{H}_r \triangleq \sum_{i=1}^n |\mathcal{H}_i| / K$ is the percentage of generated data belonging to any of the modes of real data. *Hit distance* $\mathcal{H}_d \triangleq \sum_{i=1}^n \sum_{\mathbf{x} \in \mathcal{H}_i} \epsilon_i(\mathbf{x}) / \sum_{i=1}^n |\mathcal{H}_i|$ is the averaged projection distance over all data in the hit set. Lastly, *cover error* $\mathcal{C}_e$ evaluates how well the generated data covers each mode. Essentially it computes the KL-divergence between the estimated distribution of samples in $\mathcal{PH}_i$ and the uniform distribution over $[-1, 1]^d$. Formally, it is defined as the averaged KL-divergence on $n$ modes i.e. $\mathcal{C}_e \triangleq \frac{1}{n} \sum_{i=1}^n \mathrm{KL}(\hat{p}(\cdot; \mathcal{PH}_i) \| \mathcal{U}[-1, 1]^d)$. The intuition is that if data generated is close to the ground truth distribution, they should be uniformly distributed in the square area of each mode.

**Optimization-Based v.s. Probabilistic.** The left part of Table 2 summarizes the results in terms of *hit ratio* and *hit distance*. Probabilistic methods including our algorithms and BGAN always achieve a hit ratio of 1, which means every data point generated from these models is very close to one mode of the target distribution. On the other hand, optimization based methods, both GAN and MGAN, consistently have a significantly larger hit error, and sometimes may even generate data samples that do not belong to any mode. Moreover, the data distribution generated by the optimization-based methods fits the target uniform distribution much worse than its probabilistic counterparts, which is quantitatively reflected by the cover error showed in the right side of Table 2 and visually demonstrated by the projected hit sets in Figure 2. According to the visualization, data generated by GAN or MGAN tend to be under dispersed and hardly cover the whole square region of the true mode, while data generated by probabilistic methods align much better with the ground truth distribution. We attribute this superiority to stronger exploration power in the generator space coming from the randomness in probabilistic methods.

**Bayesian GAN v.s. ProbGAN.** The incompatibility issue of BGAN with minimax-style GAN objectives theoretically derived in Section 4.2 is empirically verified in our experiments. As visualized in Figure 2, with the GAN-MM objective, BGAN is trapped in a local equilibrium and fails in capturing one mode of the true data. Besides, as shown in Table 2, BGAN with the WGAN objective achieves much poorer coverage than with other GAN objectives, while our model is much more robust to the choice of GAN objectives (consistently lower cover errors). A qualitative comparison is made in Figure 9 (in the appendix) which shows the data distribution generated by BGAN trained

---

[3] Based on the fact that the average distance between the data from two different modes is 800, we set a threshold of $\eta = 40$.

Table 2: Hit ratios ($\mathcal{H}_r$), hit distances ($\mathcal{H}_d$), cover errors ($\mathcal{C}_e$) results. Note, if the model failed to capture all the modes of real data, by definition its cover error is $\infty$. In that case, we report the averaged KL-divergence on modes captured by the model in brackets.

| | $\mathcal{H}_r$ (HIGHER IS BETTER), $\mathcal{H}_d$ (LOWER IS BETTER) | | | | $\mathcal{C}_e$ (LOWER IS BETTER) | | | |
| | GAN-MM | GAN-NS | WGAN | LSGAN | GAN-MM | GAN-NS | WGAN | LSGAN |
|---|---|---|---|---|---|---|---|---|
| GAN | 0.86, 22.6 | 0.85, 23.1 | 0.78, 26.7 | 0.74, 23.1 | 12.11 | 8.86 | 7.20 | $\infty$ (12.07) |
| MGAN | 0.82, 24.2 | 0.84, 25.5 | 0.67, 31.7 | 0.81, 23.6 | 5.46 | 6.31 | 5.00 | $\infty$ (4.25) |
| BGAN | 1.0, **5.5** | 1.0, 6.4 | 1.0, **12.1** | 1.0, 6.3 | $\infty$ (1.73) | 1.76 | 4.32 | 1.80 |
| PROBGAN-GMA | 1.0, 7.4 | 1.0, 7.7 | 1.0, 15.5 | 1.0, **5.3** | 1.84 | **1.73** | 3.01 | 1.79 |
| PROBGAN-PSA | 1.0, 5.8 | 1.0, **6.4** | 1.0, 12.5 | 1.0, 6.4 | **1.75** | 1.75 | **2.28** | **1.74** |

Table 3: Inception score and FID results on CIFAR-10. Results of each model trained with 4 different GAN objectives are all reported.

| | INCEPTION SCORES (HIGHER IS BETTER) | | | | FIDS (LOWER IS BETTER) | | | |
| | GAN-MM | GAN-NS | WGAN | LSGAN | GAN-MM | GAN-NS | WGAN | LSGAN |
|---|---|---|---|---|---|---|---|---|
| DCGAN | 6.53 | 7.21 | 7.19 | 7.36 | 35.57 | 27.68 | 28.31 | 29.11 |
| MGAN | 7.19 | 7.25 | 7.18 | 7.34 | 30.01 | 27.55 | 28.37 | 30.72 |
| BGAN | 7.21 | 7.37 | 7.26 | **7.46** | 29.87 | 24.32 | 29.87 | 29.19 |
| PROBGAN-PSA | **7.75** | **7.53** | **7.28** | 7.36 | **24.60** | **23.55** | **27.46** | **26.90** |

with WGAN objective tends to shrink. More visual illustrations under different GAN objectives are placed in Section E of the appendix.

## 5.2 NATURAL IMAGE DATASET

**Datasets.** We evaluate our method on 3 widely-adopted datasets: CIFAR-10 (Krizhevsky et al., 2010), STL-10 (Coates et al., 2011) and ImageNet (Deng et al., 2009). CIFAR-10 has 50k training and 10k test 32x32 RGB images from 10 classes: airplane, automobile, bird, cat, deer, dog, frog, horse, ship, and truck. STL-10, containing 100k 96x96 RGB images, is a more diverse dataset than CIFAR-10. ImageNet has over 1.2 million images from 1,000 classes and presents the most diverse dataset. For a fair comparison with baselines, we use the same settings as MGAN. We resize the STL-10 and ImageNet images down to 48x48 and 32x32 respectively.

**Evaluation Protocols.** We employ two common image generation metrics: *Inception Score* (Salimans et al. (2016)) and *Fréchet Inception Distance* (Heusel et al. (2017)). Inception Score computes $\exp(\mathbb{E}_{\mathbf{x}}[KL(p(y|\mathbf{x})\|p(y))])$ where $p(y|\mathbf{x})$ standards the predicted label distribution by a pre-trained Inception model (Szegedy et al., 2015) and $p(y)$ is the average of $p(y|\mathbf{x})$ over all images in the dataset. Inception Score (IS) reflects the fidelity and diversity of images and is well-correlated with human judgment (Salimans et al., 2016). However, it does not measure the similarity between the real data and the synthetic data. Therefore, as a complementary, we use Fréchet Inception Distance (FID). Specifically, it measures the Fréchet distance between two image distributions in the feature embedding space conduct by the Inception model. More details are included in the appendix

**Model Architectures.** Inspired by Hoang et al. (2018), we adapted the parameter sharing technique for both generators and discriminators. Specifically, generators (of one model) are not disjoint neural networks. They only differ at the first layer and sharing all parameters at the following layers. This tied parameter design reduced the model complexity and boost the performance. Especially for the probabilistic model such as Bayesian GAN, we observe a significant enhancement of generated image qualities compare to the original result (Figure 7 in Saatci & Wilson (2017)). More implementation details such as hyper-parameters of network layers are reported in the appendix (Section B).

**Quantitative Results.** Table 3 and Table 4 summarize the Inception scores and FIDs on the three benchmark datasets for our model and baselines. For each model, we denote the epoch delivering highest 'IS−0.1FID' as the *best epoch* and report the scores of the model at this epoch. We design this 'IS−0.1FID' principle to neutralize the discrepancy between Inception score and FID. Because Inception score and FID may not be consistent with each other, it is possible that the Inception score improves but the FID gets worse when we evaluate the model at a different checkpoint. Hence, we need a rule to pick one point of the Pareto frontier of the model performance.

Table 4: Inception score and FID results on STl-10 and ImageNet. Each model is trained with GAN-NS objective.

| DATASET | STL-10 | | IMAGENET | |
|---|---|---|---|---|
| | INCEPTION SCORES | FIDs | INCEPTION SCORES | FIDs |
| DCGAN | $8.05 \pm 0.101$ | 51.01 | $7.66 \pm 0.113$ | 48.99 |
| MGAN | $8.72 \pm 0.096$ | 51.56 | $7.77 \pm 0.108$ | 45.75 |
| BGAN | $8.84 \pm 0.100$ | 47.35 | $8.52 \pm 0.075$ | 29.68 |
| PROBGAN-PSA | $\mathbf{8.87} \pm 0.095$ | **46.74** | $\mathbf{8.57} \pm 0.073$ | **27.69** |

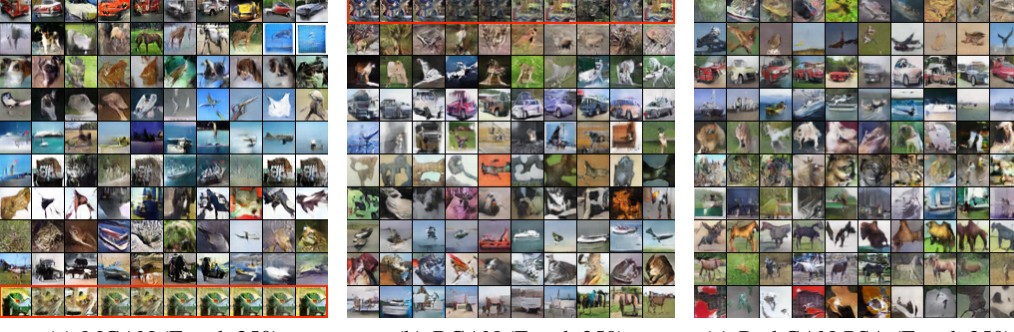

(a) MGAN (Epoch 250)      (b) BGAN (Epoch 250)      (c) ProbGAN-PSA (Epoch 250)

Figure 3: Images generated by MGAN, BGAN and our model trained on CIFAR 10 with GAN-NS objective. The tenth generator of MGAN (Figure 3(a)) and the first of BGAN (Figure 3(b)) collapse while generators of our method all work well. DCGAN (Figure 10 in the appendix) also presents 'single generator collapse' issue. Note that, mode collapse also happens when baseline models trained with other GAN objectives.

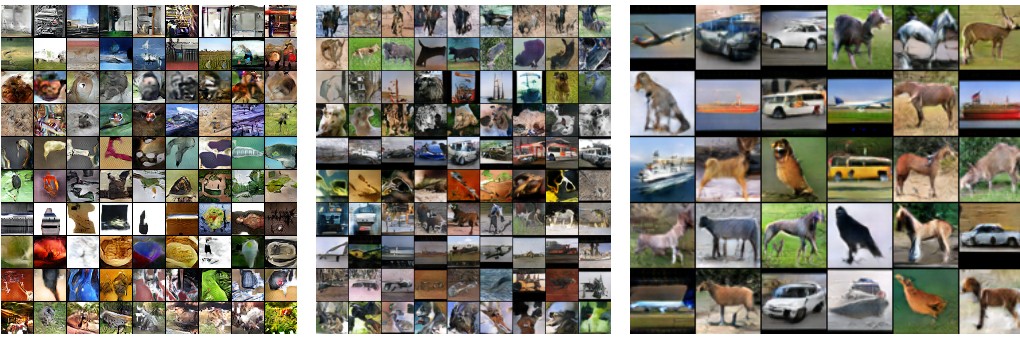

(a) ImageNet (randomly picked)   (b) STL-10 (randomly picked)     (c) STL-10 (cherry-picked)

Figure 4: Images generated by ProbGAN trained on ImageNet (left) and STL-10 (middle, right). Figure 4(c) are cherry-picked synthetic images on STL-10.

Overall, our proposed ProbGAN outperforms the baselines on all three benchmark image datasets. Furthermore, according to results on CIFAR-10, our model achieves the best performance under all GAN objectives. Generally, probabilistic methods achieve better scores than optimization based methods, which indicates that injecting stochasticity into GAN training helps generate more multi-modal images. Note that the performance gap increases as the dataset gets more diverse. Specifically, when using GAN-NS objective, our model improves FID by 4.00 on CIFAR-10 (Table 3) in comparison to MGAN. While the FID improvements are 4.82 and 18.06 on STL-10 and ImageNet (Table 4), respectively.

Besides, we note that Bayesian GAN has a significant performance drop when accompanied by min-max style GAN objectives, which provides another empirical evidence for our theory analysis in Section 4.2. By contrast, ProbGAN fits any GAN objectives and constantly performs well.

**Qualitative Results.** Figure 3 displays the samples randomly generated by the baselines and our ProbGAN. Each row in the figure contains samples of one learned generator. In Figure 3 and Figure 10 (in the appendix), all the three baselines noticeably suffer from mode collapse. Indeed, almost in every training trial, one or two generators of the baseline models degenerate during the training. Hoang et al. (2018) already notice that mode collapse in one of the generators could happen after a

long training procedure (around 250 epochs). Our experiment shows Bayesian GAN also have this issue. However, ProbGAN is robust to mode collapse. Visual results for the entire ablation study on CIFAR-10 are included in Section F of the appendix.

Figure 4(a) and Figure 4(b) display images generated by our model trained on ImageNet and STL-10. Each row in these two figures contains generated images from one generator. The results show that our method is robust to 'single generator mode collapse' in both STL-10 and ImageNet. We also include cherry-picked results on STL-10 to exhibit the capability of our model to generate visually appealing images with complex details while improving the robustness against mode collapse.

## 6 DISCUSSION

In this paper, we propose ProbGAN, a novel probabilistic modelling framework for GAN. From the perspective of Bayesian Modelling, it contributes a novel likelihood function establishing a connection to existing GAN models and a novel prior stabilizing the inference process. We also design scalable and asymptotically correct inference algorithms for ProbGAN. In the future work, we plan to extend the proposed framework to non-parametric Bayesian modelling and investigate more theoretical properties of GANs in the probabilistic modelling context.

Developing Bayesian generalization for deep learning models is not a recent idea and happens in many fields other than generative models such as adversarial training (Ye & Zhu, 2018) and Bayesian neural networks (Wang et al., 2016a). By this work, we emphasize the importance of going beyond the intuition and understanding the theoretical behavior of the Bayesian model. We hope that our work helps inspire continued exploration into Bayesian deep learning (Wang & Yeung, 2016) from a more rigorous perspective.

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

## A    OMITTED PROOFS

**Theorem 1.** This theorem is general and holds when the GAN objective and the discriminator space have symmetry. The symmetry of GAN objective means its functions $\phi_1$ and $\phi_2$ satisfy that $\exists c \in \mathbb{R}, \forall x \in \mathbb{R}, \phi_1(x) \equiv \phi_2(c - x)$. While the symmetry of discriminator space $\Theta_d$ indicates that for any $\theta_d \in \Theta_d$, there is a $\theta'_d \in \Theta_d$ such that $D(\mathbf{x}; \theta_d) \equiv c - D(\mathbf{x}; \theta'_d)$. Note that the symmetry condition are very weak, first it holds for all the common choices of GAN objectives such as those listed in Table 1. Second, it holds for neural network which is the most common parameterization for discriminator in practice.

*Proof.*

$$q_d^*(\theta_d) \propto \exp(\mathbb{E}_{\mathbf{x} \sim p_{data}}[\phi_1(D(\mathbf{x}; \theta_d))] + \mathbb{E}_{\mathbf{x} \sim p_{model}^*}[\phi_2(D(x; \theta_d))])$$
$$= \exp(\mathbb{E}_{\mathbf{x} \sim p_{data}}[\phi_1(D(\mathbf{x}; \theta_d)) + \phi_2(D(x; \theta_d))])$$
$$= \exp(\mathbb{E}_{\mathbf{x} \sim p_{data}}[\phi_2(c - D(\mathbf{x}; \theta_d)) + \phi_1(c - D(x; \theta_d))]) \quad (11)$$
$$= \exp(\mathbb{E}_{\mathbf{x} \sim p_{data}}[\phi_2(D(\mathbf{x}; \mathcal{S}(\theta_d))) + \phi_1(D(x; \mathcal{S}(\theta_d)))])$$
$$\Rightarrow q_d^*(\theta_d) = q_d^*(\mathcal{S}(\theta_d)),$$

$$D^*(\mathbf{x}) = \mathbb{E}_{\theta_d \sim q_d^*} D(\mathbf{x}; \theta_d) = \int_{\theta_d \in \Theta_d} q_d^*(\theta_d) D(\mathbf{x}; \theta_d) d\theta_d$$
$$= \frac{1}{2} \left( \int_{\theta_d \in Td} q_d^*(\theta_d) D(\mathbf{x}; \theta_d) d\theta_d + \int_{\theta_d' = \mathcal{S}(\theta_d) \in Td} q_d^*(\theta_d') D(\mathbf{x}; \theta_d') d\theta_d' \right)$$
$$= \frac{1}{2} \int_{\theta_d \in \Theta_d} q_d^*(\theta_d) (D(\mathbf{x}; \theta_d) + D(\mathbf{x}; \mathcal{S}(\theta_d))) d\theta_d \quad (12)$$
$$= \frac{1}{2} \int_{\theta_d \in \Theta_d} q_d^*(\theta_d) \cdot c \cdot d\theta_d = \frac{c}{2}.$$

Eqn. 11 and Eqn. 12 prove that $q_d^*(\theta_d) \propto \exp(\mathcal{J}_d(\theta_d; p_{model}^*))$ satisfies $\mathbb{E}_{\theta_d \sim q_d^*} D(\cdot; \theta_d) \equiv \frac{c}{2}$. Note that, in the above equations, $\mathcal{S}(\theta_d)$ denotes the symmetric discriminator of $\theta_d$ satisfying $D(\mathbf{x}; \mathcal{S}(\theta_d)) \equiv c - D(\mathbf{x}; \theta_d)$. Hence according Eqn. 13, we know $\mathcal{J}_g(\theta_g; D^*)$ is a constant.

$$\mathcal{J}_g(\theta_g; D^*) = \mathbb{E}_{\mathbf{x} \sim p_{gen}(\cdot; \theta_g)}[\phi_3(D^*(\mathbf{x}))] = \phi_3\left(\frac{c}{2}\right), \quad \forall \mathbf{x} \in \mathcal{X}. \quad (13)$$

Thus the generator distribution will not change based on the dynamics in Eqn. 2 since $q_g^*(\theta_g) \propto \exp\{\mathcal{J}_g(\theta_g; D^*)\} q_g^*(\theta_g)$. $\qquad\square$

## Corollary 1

*Proof.*

$$\log q_g^{(t+1)}(\theta_g) = \frac{1}{J_g J_d} \sum_{i=1}^{J_g} \sum_{k=1}^{J_d} \log p(\theta_g | \mathbf{z}^{(i)}, \theta_d^k) \quad (14)$$

$$\log q_g^{(t+1)}(\theta_g) \simeq \mathbb{E}_{\theta_d \sim q_d(\theta_d)} \mathcal{J}_g(\theta_g; \theta_d) + \log p(\theta_g | \alpha_g) \quad (15)$$

Algorithm 1 from the original paper (Saatci & Wilson, 2017) implies that Eqn. 14 where $\theta_d^k \sim q_d^{(t)}(\theta_d)$ are Monte Carlo samples of discriminator and every $\mathbf{z}^{(i)}$ is a mini-batch containing $n_g$ noise samples. Be definition $p(\theta_g | \mathbf{z}^{(i)}, \theta_d^k) \propto \prod_{j=1}^{n_g} D(G(\mathbf{z}_j^{(i)}; \theta_g); \theta_d^k) p(\theta_g | \alpha_g)$ (Eqn. 1 in the original paper), each term[4] $\log p(\theta_g | \mathbf{z}^{(i)}, \theta_d^k) = \sum_{j=1}^{n_g} \phi_3(D(G(\mathbf{z}_j^{(i)}; \theta_g); \theta_d^k)) + \log p(\theta_g | \alpha_g)$. Hence, the total summation is a Monte Carlo approximation of the expectation in the right side of Eqn. 15. The same derivation can be done to $q_d^{(t+1)}$. Together, we get Corollary 1. $\qquad\square$

## Lemma 1

*Proof.*

$$p(x|y) = \alpha(y) \exp\{L(x, y)\} q_x(x),$$
$$p(y|x) = \beta(x) \exp\{-L(x, y)\} q_y(y),$$
$$\implies p(x, y)^2 = p(x|y)p(y) \times p(y|x)p(x)$$
$$= p(x)p(y)\alpha(y)\beta(x)q_x(x)q_y(y) \quad (16)$$
$$\implies X, Y \text{ are independent.}$$
$$\implies p(x) = p(x|y)$$
$$\implies L(x, y) = \log p(x) - \log q_x(x) - \log \alpha(y)$$
$$\implies L(x, y) \text{ is decomposable.}$$

---

[4]Note that in BGAN paper, the GAN objective is GAN-NS. Thus $\phi_3$ equals $\log(\cdot)$.

where $\alpha(y) = (\int \exp\{L(x,y)\}q_x(x)dx)^{-1}$ and $\beta(x) = (\int \exp\{-L(x,y)\}q_y(y)dy)^{-1}$. $\qquad\square$

## B  EXPERIMENT DETAILS

### B.1  NATURAL IMAGE DATASET

For MGAN results, we adopt the official Tensorflow implementation[5]. While, for our model and other baselines, we implement them in PyTorch (Paszke et al., 2017).

**Remark on Inception score and FID**. Barratt & Sharma (2018) point out that Inception score is sensitive to the inception model used and the number of data splits in the computation. This is also true for Fréchet Inception Distance (FID). We find that the FID computed by a PyTorch Inception model[6] is much lower than the FID given by a Tensorflow model[7].

In our experiments, to facilitate a fair comparison with prior work, we compute Inception score and FID using the Tensorflow Inception model. We adopt the official Tensorflow implementation for FID to compute both Inception score and FID. We will release our evaluation code soon.

**Model architecture**: In our experiments, each model is trained with 10 generators. As for discriminator, DCGAN and MGAN have one discriminator while probabilistic models (BGAN and ours) have 4 discriminators (i.e. 4 Monte Carlo samples from discriminator distribution).

The neural network structures are the same as MGAN. As reported in Table 4,5,6 in Section C.2 of the original MGAN paper. Briefly, the structures are the following.

Generator architecture has four (or five) deconvolution layers (kernel size 4, stride 2) with the following input, hidden feature-maps, output size: 100x1x1 $\to$ 512x4x4 $\to$ 256x8x8 $\to$ 128x16x16 $\to$ 3x32x32. Every deconvolution layer is followed by batch-normalization layer and Relu activation except for the last deconvolution layer who is followed by Tanh activation.

Discriminator architecture has four (or five) convolution layers (kernel size 5, stride 2) with the following input, hidden feature-maps, output size: 3x32x32 $\to$ 128x16x16 $\to$ 256x8x8 $\to$ 512x4x4 $\to$ 1x1x1. Batch-normalization is applied to each layer except the last one. Activations are leaky-ReLU.

**Training hyperparameters**: All models are optimized by Adam(Kingma & Ba, 2014) with a learning rate of $2 \times 10^4$. For probabilistic methods, the SGHMC noise factor is set as $3 \times 10^2$. Following the configuration in MGAN, the batch size of generators and discriminators are 120 and 64. Note that, since probabilistic model has 10 generator Monte Carlo samples and 4 discriminators, indeed batch size for every generator and discriminator is 12 and 16 respectively.

### B.2  HIGH-DIMENSIONAL MULTI-MODAL SYNTHETIC DATASET

**Model Architecture**: Each generator or discriminator is a three layer perceptron. For the generator, the dimensions of input, hidden layer and output are 10, 1000, and 100 respectively. For the discriminator, the dimensions of input, hidden, output layers are 100, 1000, and 1. All activation functions are leaky ReLU(Maas et al., 2013).

**Training hyperparameters**: All models are optimized by Adam (Kingma & Ba, 2014) with a learning rate of $10^{-4}$. For probabilistic methods, the SGHMC noise factor ($\alpha$ in algorithm 1) is set as $10^{-1}$.

## C  INFERENCE ALGORITHM

Stochastic Gradient Hamiltonian Monte Carlo (Chen et al., 2014) is a gradient based MCMC sampling method. It use noise gradient estimation of potential function to generate sample from a given distribution. Our inference algorithm (algorithm 1) is based on this brilliant algorithm.

---

[5]MGAN Code: https://github.com/qhoangdl/MGAN

[6] https://github.com/mseitzer/pytorch-fid

[7]https://github.com/bioinf-jku/TTUR

**Input:** Initial Monte Carlo samples of $\{\theta_{d,m}^{(0)}\}_{m=1}^{M_d}$ and $\{\theta_{g,m}^{(0)}\}_{m=1}^{M_g}$, learning rate $\eta$, SGHMC noise factor $\alpha$, number of updates in SGHMC procedure $L$, number of updating iterations $T$.

**for** $t = 1$ **to** $T$ **do**
    **for** $m = 1$ **to** $M_d$ **do**
        $\theta_{d,m} \leftarrow \theta_{d,m}^{(t)}$
        **for** $l = 1$ **to** $L$ **do**
            $\mathbf{v} \leftarrow (1 - \alpha)\mathbf{v} + \eta \nabla_{\theta_d} \log q_d^{(t+1)}(\theta_{d,m}) + \mathbf{n}; \mathbf{n} \sim \mathcal{N}(0, 2\alpha\eta I)$
            $\theta_{d,m} \leftarrow \theta_{d,m} + \mathbf{v}$
        **end for**
        $\theta_{d,m}^{(t+1)} \leftarrow \theta_{d,m}$
    **end for**
    **for** $m = 1$ **to** $M_g$ **do**
        $\theta_{g,m} \leftarrow \theta_{g,m}^{(t)}$
        **for** $l = 1$ **to** $L$ **do**
            $\mathbf{v} \leftarrow (1 - \alpha)\mathbf{v} + \eta \nabla_{\theta_g} \log q_g^{(t+1)}(\theta_{g,m}) + \mathbf{n}; \mathbf{n} \sim \mathcal{N}(0, 2\alpha\eta I)$
            $\theta_{g,m} \leftarrow \theta_{g,m} + \mathbf{v}$
        **end for**
        $\theta_{g,m}^{(t+1)} \leftarrow \theta_{g,m}$
    **end for**
**end for**

**Algorithm 1:** Our Adapted SGHMC Inference Algorithm

Table 5: Configuration of four models in comparison.

| Model | Likelihood | Prior |
|---|---|---|
| A | expectation of objective value | Gaussian: $p_g(\theta_g) \propto \mathcal{N}(0,1), p_d(\theta_d) \propto \mathcal{N}(0,1)$ |
| B | objective value of expectation | Gaussian: $p_g(\theta_g) \propto \mathcal{N}(0,1), p_d(\theta_d) \propto \mathcal{N}(0,1)$ |
| C | expectation of objective value | ProbGAN: $p_g(\theta_g) \propto q_g^{(t)}(\theta_g), p_d(\theta_d) \propto 1$ |
| D | objective value of expectation | ProbGAN: $p_g(\theta_g) \propto q_g^{(t)}(\theta_g), p_d(\theta_d) \propto 1$ |

## D   Toy experiment on categorical distribution

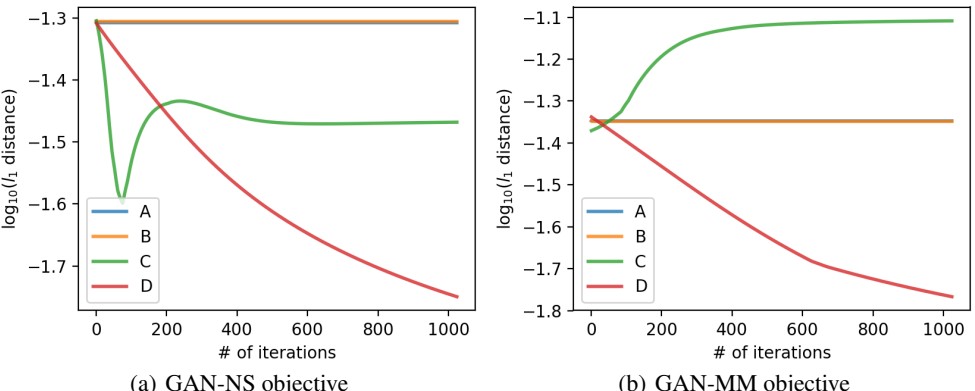

(a) GAN-NS objective        (b) GAN-MM objective

Figure 5: The $l_1$ distances between model generated data distribution and target data distribution at each iteration step. Results of model A, B, C, D are denoted by different colors.

**Setup.** In this toy example, we consider the case where $\mathcal{X}$, $\Theta_g$, and $\Theta_d$ are all finite sets, specifically $\mathcal{X} = \{\mathbf{x}_1, \cdots, \mathbf{x}_N\}$, $\Theta_g = \{\theta_g^1, \cdots, \theta_g^{N_g}\}$, $\Theta_d = \{\theta_d^1, \cdots, \theta_d^{N_d}\}$. The target data distribution is a categorical distribution $\mathrm{Cat}(\lambda_{1:N})$ where $\lambda_i = p_{data}(\mathbf{x}_i)$ is the probability of generating data $\mathbf{x}_i$. Generator $G_i$ generates data following the categorical distribution $p_{data}(\mathbf{x}; \theta_g^i) = \mathrm{Cat}(\alpha_{1:N}^i)$.

Further, the probability distributions of generator and discriminator are categorical distributions $q_g(\theta_g) = \text{Cat}(\beta_{1:N_g})$ and $q_d(\theta_d) = \text{Cat}(\gamma_{1:N_d})$.

In practice, we set $N = 10, N_g = 20, N_d = 100$. In each experiment trial, target data distribution $\lambda_{1:N}$ and data distributions for each generator $\alpha_{1:N}^i$ are fixed and initialized by the following categorical distribution generating procedure. We first independently sample from the uniform distribution $\mathcal{U}[0,1]$ to get $\{\tilde{\lambda}_j\}_{j=1}^N$ and then normalize them to get a categorical distribution $\lambda_j = \frac{\tilde{\lambda}_j}{\sum_{j=1}^N \tilde{\lambda}_j}$.

For the discriminators, their function values are randomly generated from a uniform distribution, i.e. $\{D(\mathbf{x}_i; \theta_d^j)\}_{i=1,j=1}^{N,N_d} \sim \mathcal{U}[0,1]$.

**Models in comparison.** As listed in Table 5, we compare four models with different pairs of likelihood and prior. There are two choices of likelihood, *expectation of objective value* and *objective value of expectation*. Mathematically, *expectation of objective value* likelihood has the following formula:

- Generator likelihood $l_g(\theta_g)$ of observing discriminator distribution $q_d(\theta_d)$ is proportion to $exp\{\mathbb{E}_{\theta_d \sim q_d(\theta_d)} \mathcal{J}_g(\theta_g; \theta_d)\}$.

- Discriminator likelihood $l_d(\theta_d)$ of observing generator distribution $q_g(\theta_g)$ is proportion to $exp\{\mathbb{E}_{\theta_g \sim q_g(\theta_g)} \mathcal{J}_d(\theta_d; \theta_g)\}$.

While the *objective value of expectation* likelihood is different as follows:

- Generator likelihood $l_g(\theta_g)$ of observing discriminator distribution $q_d(\theta_d)$ is proportion to $exp\{\mathcal{J}_g(\theta_g; \mathbb{E}_{\theta_d \sim q_d(\theta_d)} D(\cdot; \theta_d))\}$.

- Discriminator likelihood $l_d(\theta_d)$ of observing generator distribution $q_g(\theta_g)$ is proportion to $exp\{\mathcal{J}_d(\theta_d; \mathbb{E}_{\theta_d \sim q_d(\theta_d)} p_{gen}(\cdot; \theta_g))\}$.

Thus model A indeed is the design of Bayesian GAN, while model D is our proposed model. Model B and C are introduced to conduct the ablation study.

**Metric.** We employ $l_1$ distance for evaluation which can be directly computed on categorical distributions as follows.

$$\mathcal{D}_{l_1}(p_{data}, p_{model}) = \sum_{\mathbf{x} \in \mathcal{X}} |p_{data}(\mathbf{x}) - p_{model}(\mathbf{x})|. \tag{17}$$

**Evaluation.** In the categorical distribution settings, all the likelihood, prior and posterior computing can be done analytically. For each model, we update the generator and discriminator distributions iteratively and monitor the distance between the target data distribution and the data distribution generated by the model. We experiment with two different choices of GAN objectives, GAN-MM and GAN-NS. Note GAN-MM is a minimax-style objective.

**Result.** Figure 5 shows how $l_1$ distance changes as the number of updating iterations increases. As we can see, model A and model B are easily and quickly trapped into bad local minima, indicating the convergence issue caused by non-informative prior. Interestingly, model C presents bifurcate results when accompanied by different GAN objectives. Its abnormal behavior in the setting of using GAN-MM objective indicates that the *expectation of objective value* likelihood used in Bayesian GAN does not fit the minimax-style GAN objectives. Finally, our ProbGAN converges the fastest (towards the global optima) and is robust to all GAN objectives.

## E    A FULL VISUALIZATION OF PROJECTED HIT SETS

In this section, we shows projected hit sets for all models under different GAN objectives.

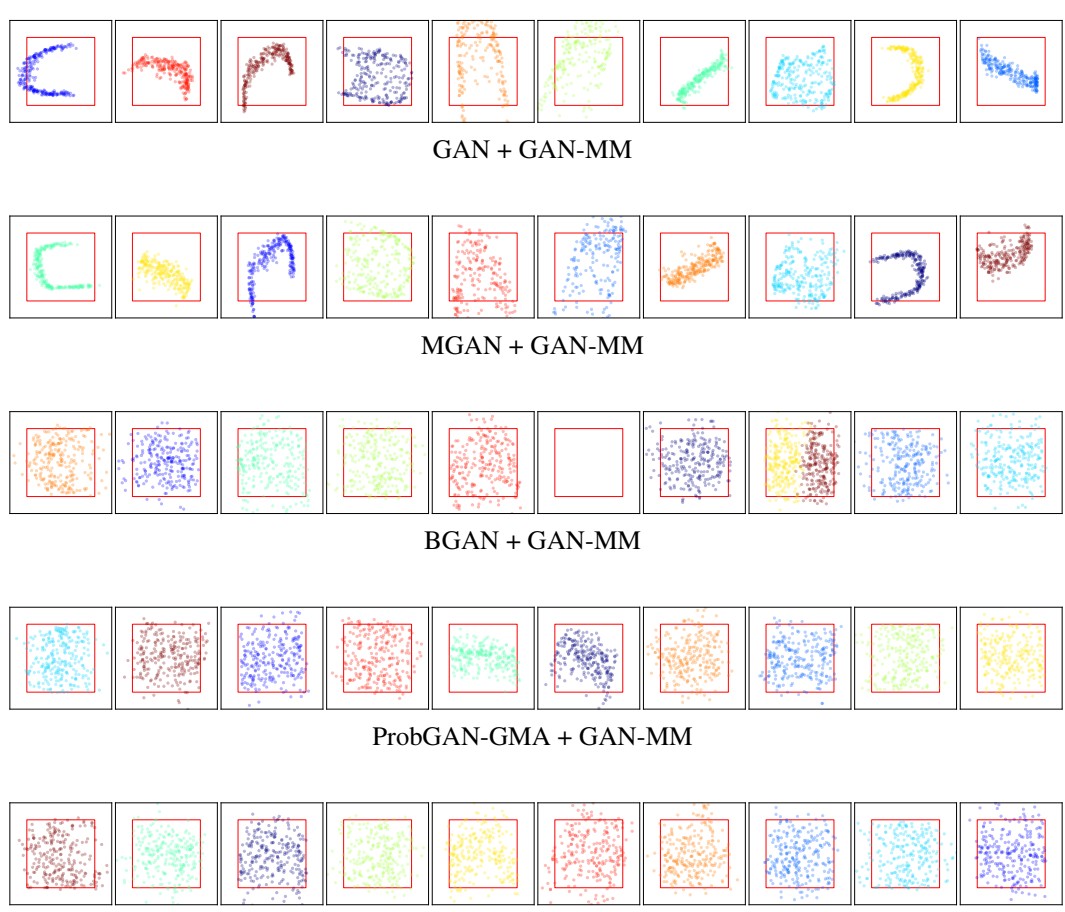

Figure 6: Visualization of the projected hit sets of all models trained with GAN-MM objective. The top two rows show the results of optimization based methods.The bottom three rows present probabilistic method results. In each row, projected hit sets for each mode are plotted in different panels. The red boxes in each panel indicate the region $\mathcal{U}[-1, 1]^2$ where the target data uniformly distributed. The data points produced by different generators of a model is painted with different colors.

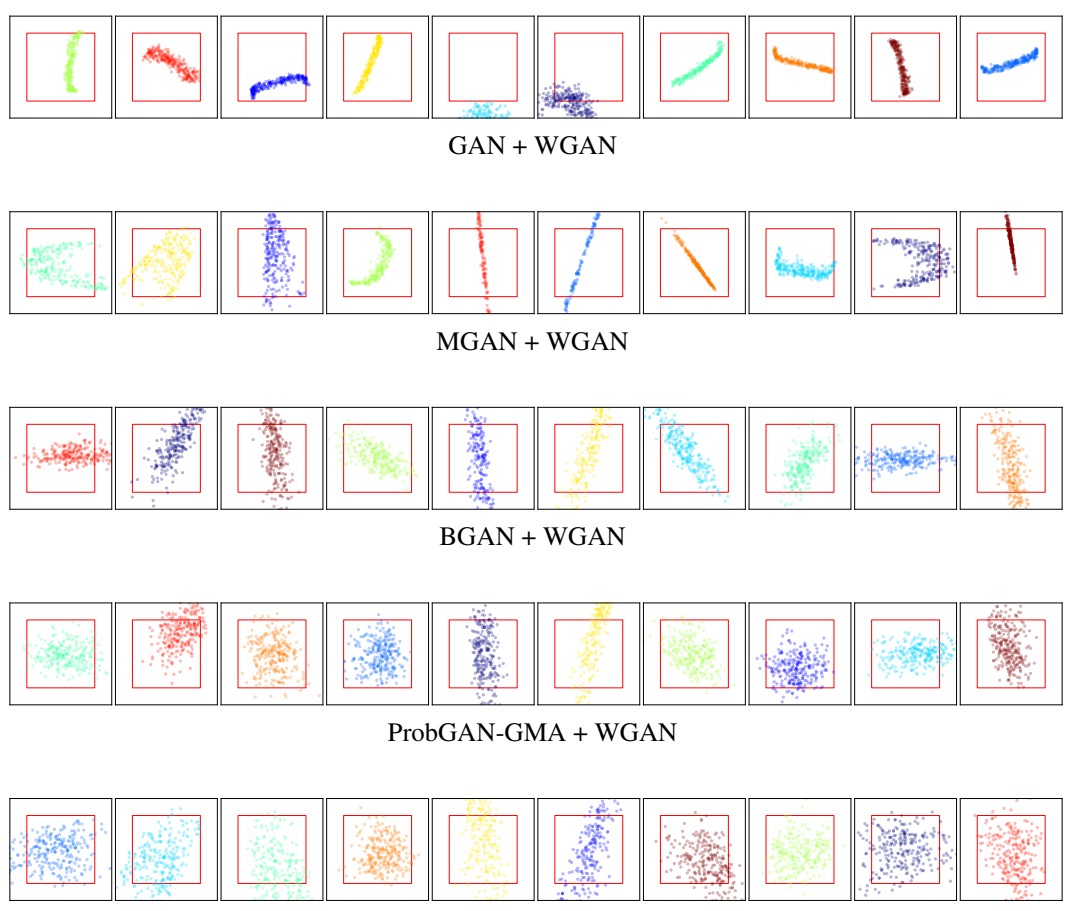

Figure 7: Visualization of the projected hit sets of different models trained with the WGAN objective. As we can see, training the WGAN objective leads to much worse performance for both optimization-based methods and BGAN. On the other hand, our methods are robust to the choice of different GAN objectives and do not suffer from significant performance drop when using the WGAN objective.

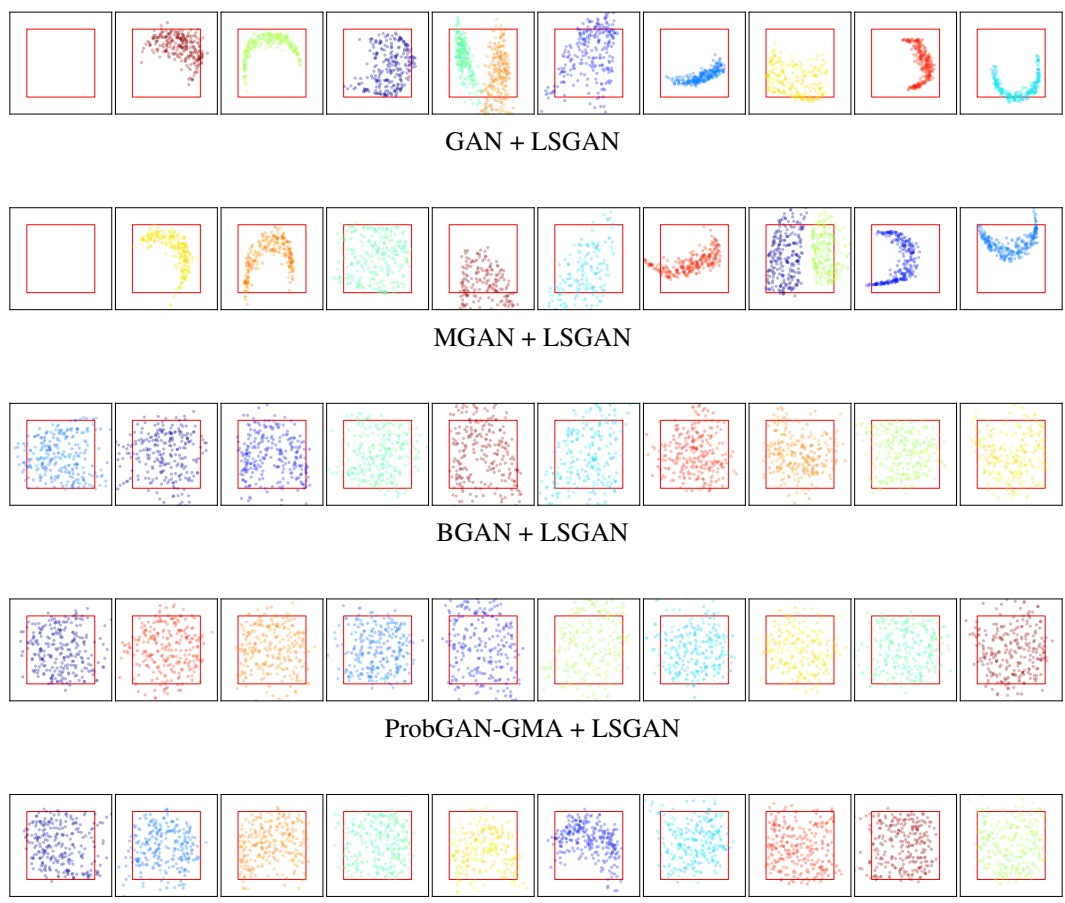

Figure 8: Visualization of the projected hit sets of different models trained with the LSGAN objective. Three probabilistic models performs perfectly in this case, while both the two optimization-based methods miss one mode of the true distribution. This experiment illustrates that although MGAN employs an additional classifier to force the data generated by different generators to be disjoint, it still suffers from mode collapsing problem. This is because in MGAN, generators may still generate disjoint data samples in the same mode and fail in capturing other modes.

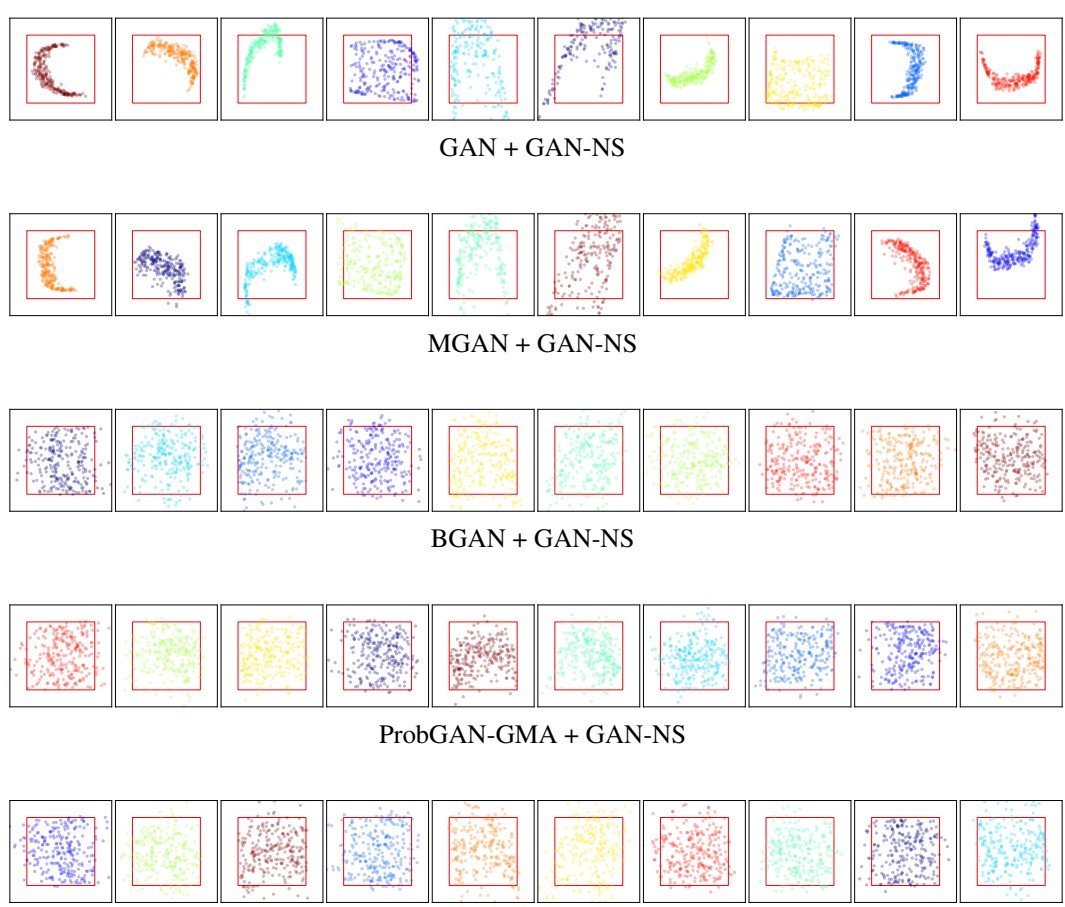

Figure 9: Visualization of the projected hit sets of different models trained with the GAN-NS objective. All the models succeed in fitting each mode of true distribution with one of their generator. Specifically, three probabilistic models generate data almost perfectly covering the ground-truth 'squares' while the optimization-based methods have difficulty covering the whole 'squares' and tend to yield under-dispersed data distributions. Note that since the GAN-NS objective is not in a min-max style, the success of BGAN is expected.

# F   MORE CIFAR-10 GENERATED IMAGE RESULTS

In this section, we shows images generated by all models trained on CIFAR-10 under different GAN objectives.

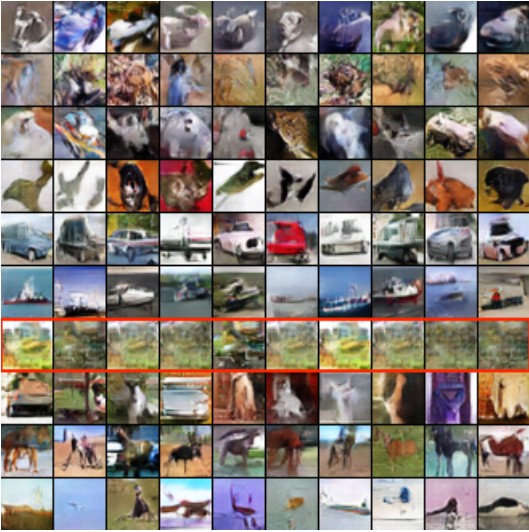

Figure 10: Images generated by **DCGAN** trained on CIFAR-10 with **GAN-NS** objective at epoch 250. Redbox indicates the collapsed generator.

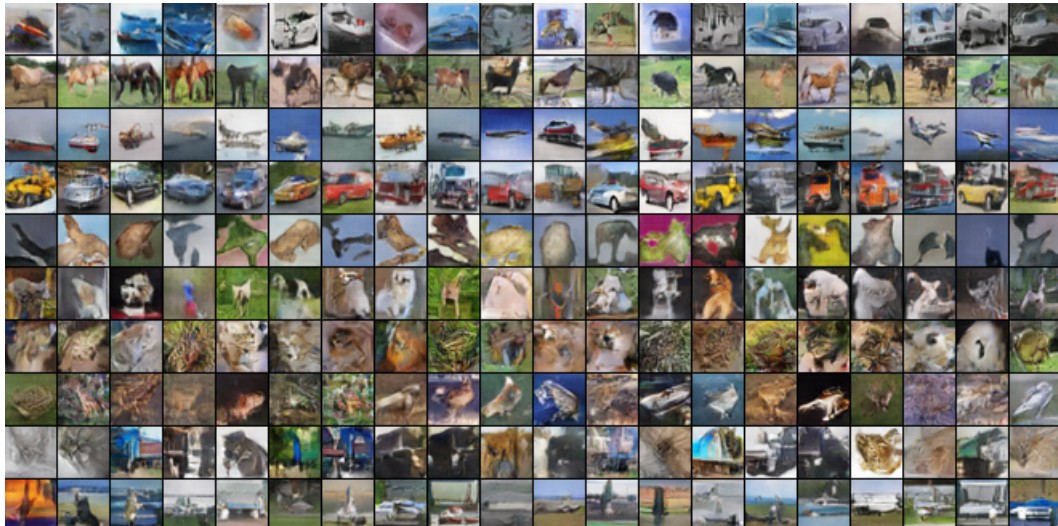

Figure 11: Images generated by **ProbGAN-PSA model** trained on CIFAR-10 with **GAN-NS** objective.Images in different rows are generated by different generators.

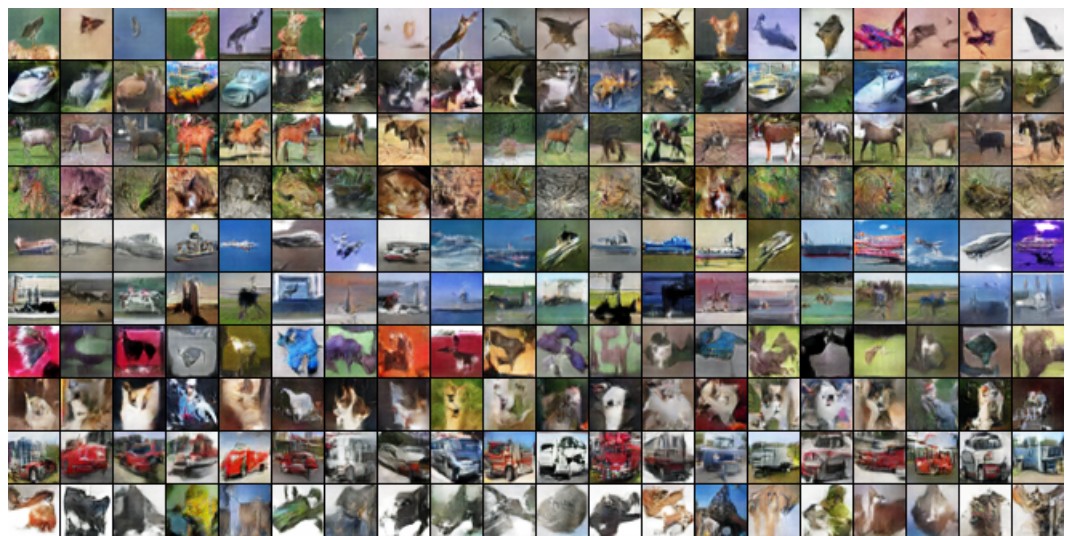

Figure 12: Images generated by **ProbGAN-PSA model** trained on CIFAR-10 with **LSGAN** objective.Images in different rows are generated by different generators.

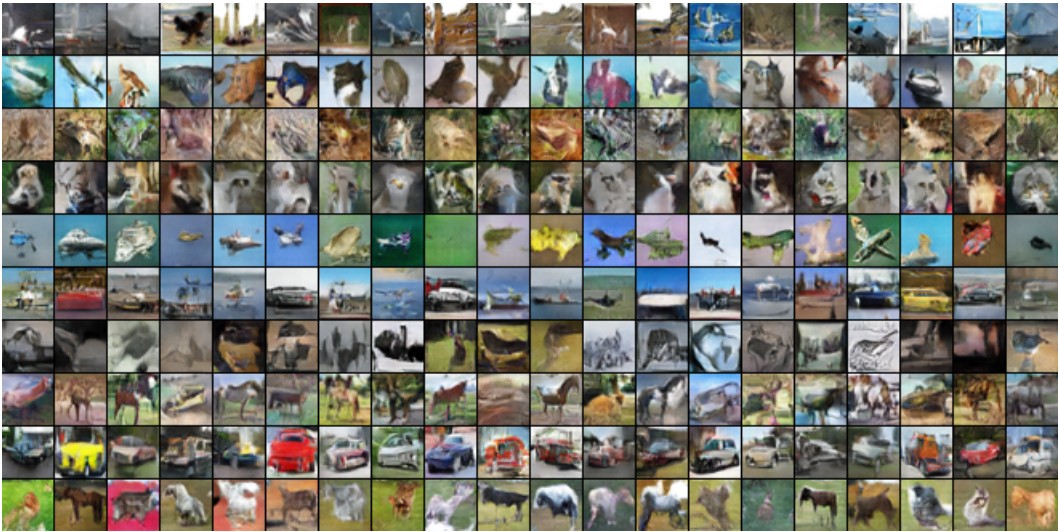

Figure 13: Images generated by **ProbGAN-PSA model** trained on CIFAR-10 with **GAN-MM** objective.Images in different rows are generated by different generators.

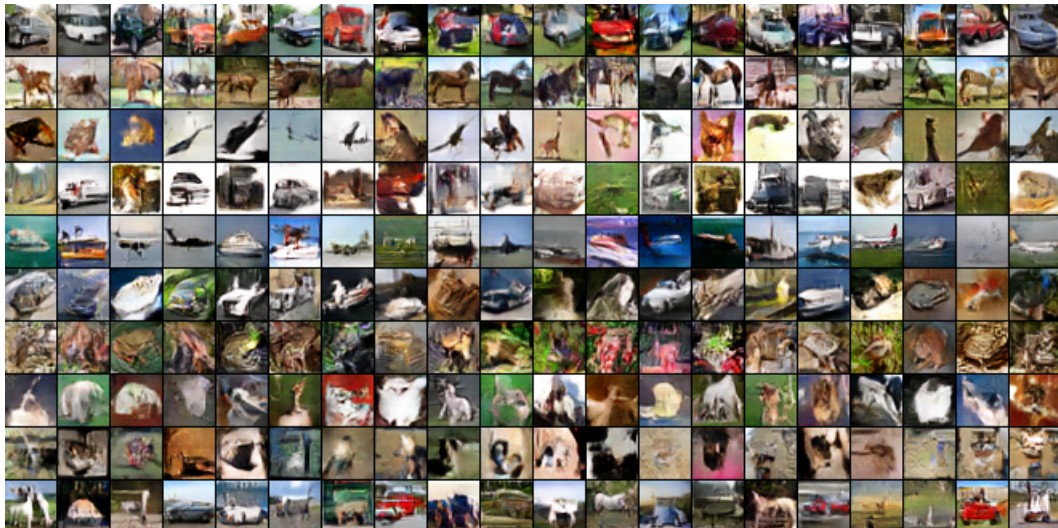

Figure 14: Images generated by **ProbGAN-PSA model** trained on CIFAR-10 with **WGAN** objective.Images in different rows are generated by different generators.

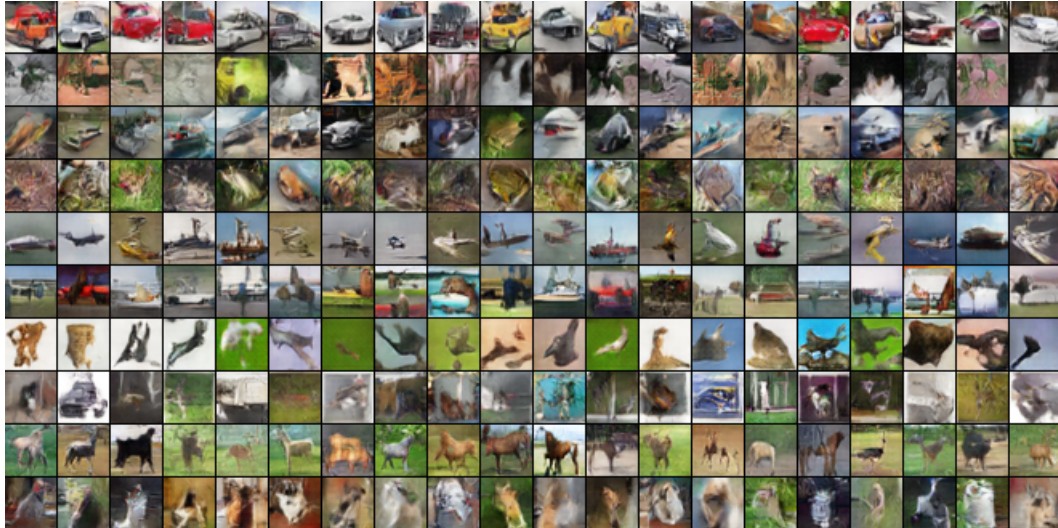

Figure 15: Images generated by **Bayesian GAN** trained on CIFAR-10 with **GAN-NS** objective.Images in different rows are generated by different generators.

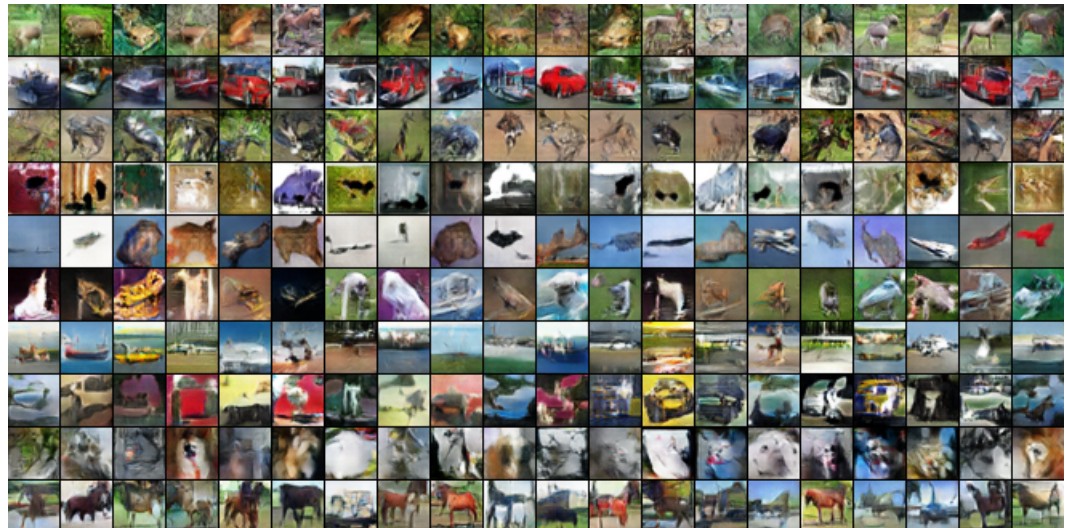

Figure 16: Images generated by **Bayesian GAN** trained on CIFAR-10 with **LSGAN** objective.Images in different rows are generated by different generators.

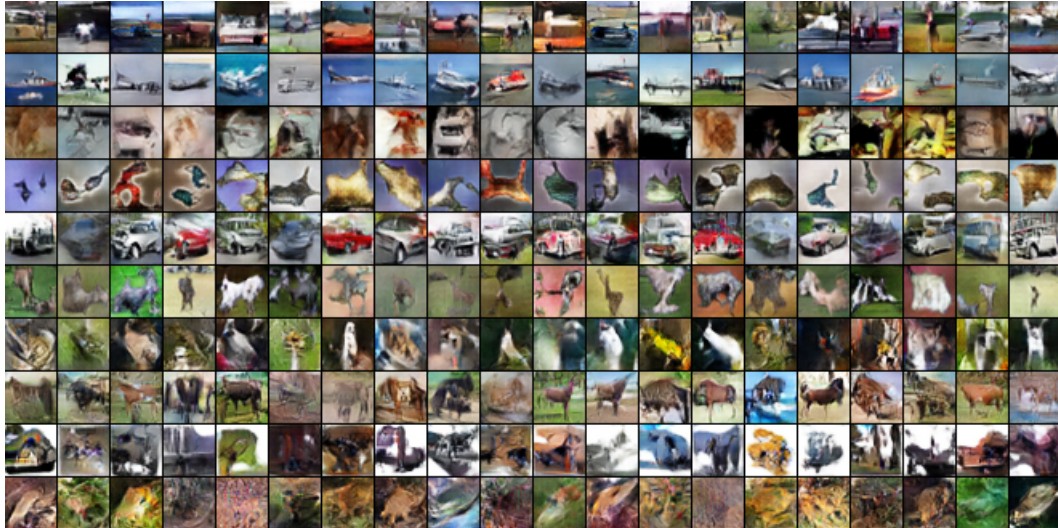

Figure 17: Images generated by **Bayesian GAN** trained on CIFAR-10 with **GAN-MM** objective.Images in different rows are generated by different generators.

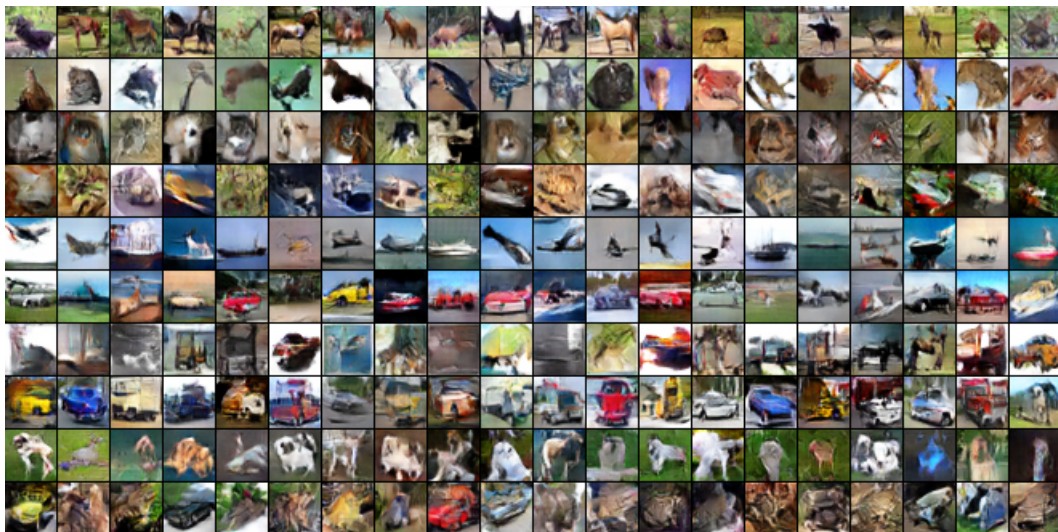

Figure 18: Images generated by **Bayesian GAN** trained on CIFAR-10 with **WGAN** objective.Images in different rows are generated by different generators.

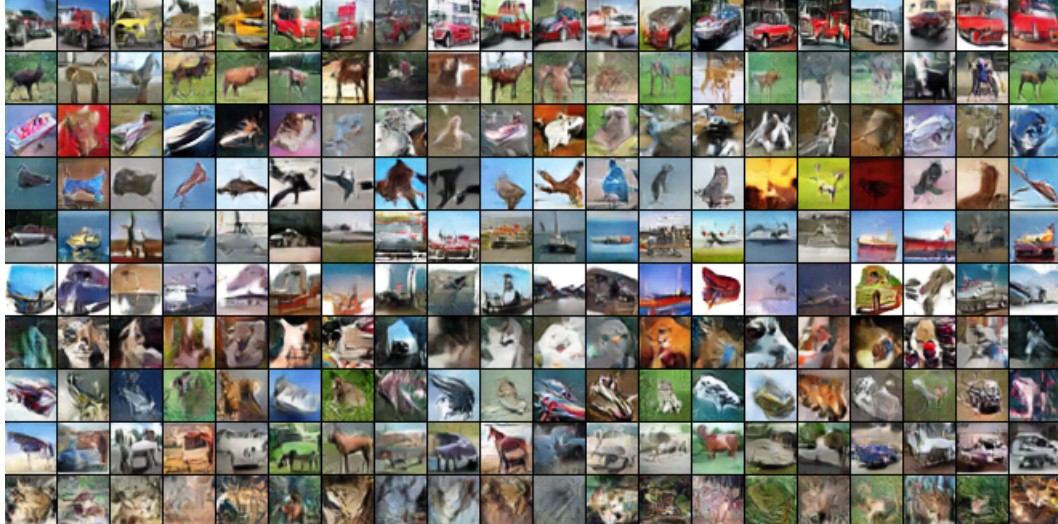

Figure 19: Images generated by **MGAN** trained on CIFAR-10 with **GAN-NS** objective.Images in different rows are generated by different generators.

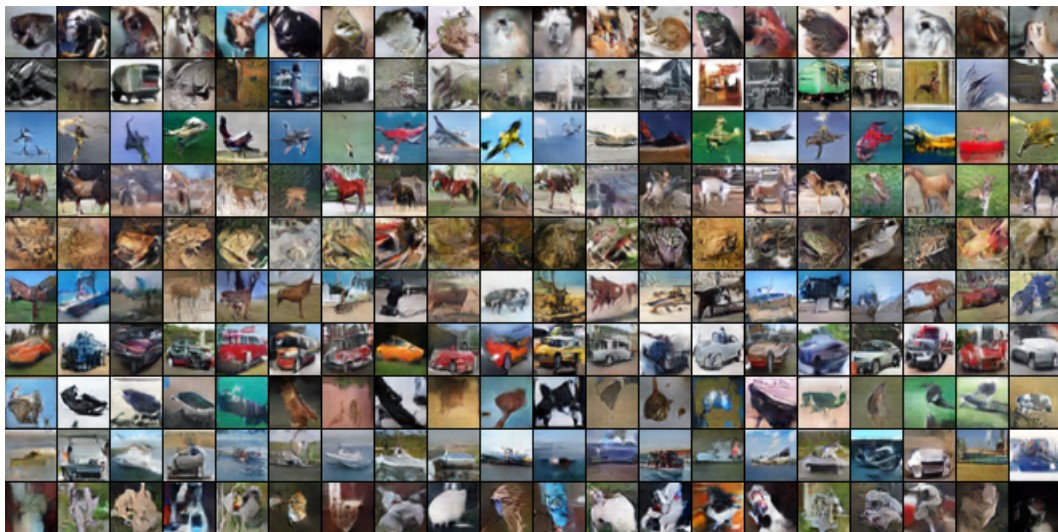

Figure 20: Images generated by **MGAN** trained on CIFAR-10 with **LSGAN** objective.Images in different rows are generated by different generators.

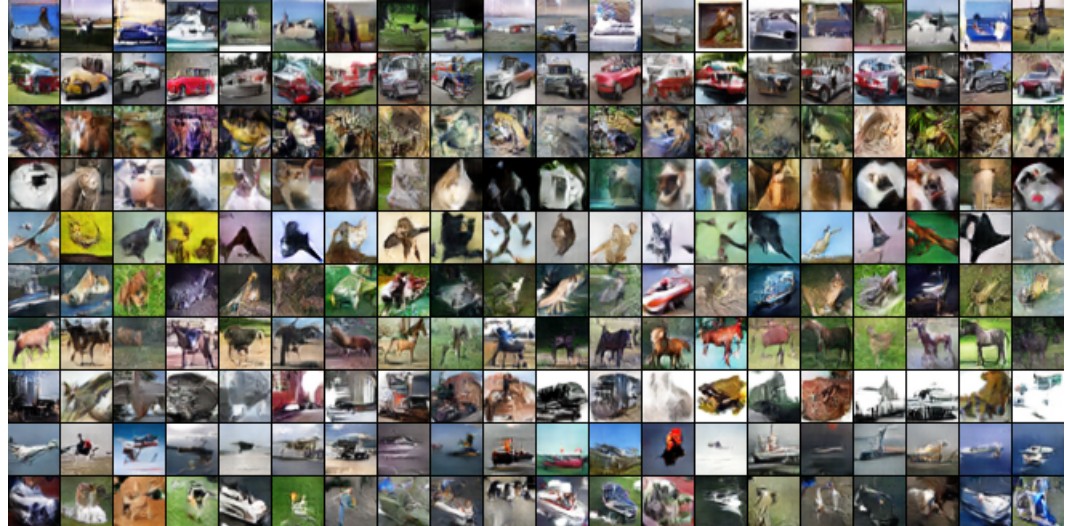

Figure 21: Images generated by **MGAN** trained on CIFAR-10 with **GAN-MM** objective.Images in different rows are generated by different generators.

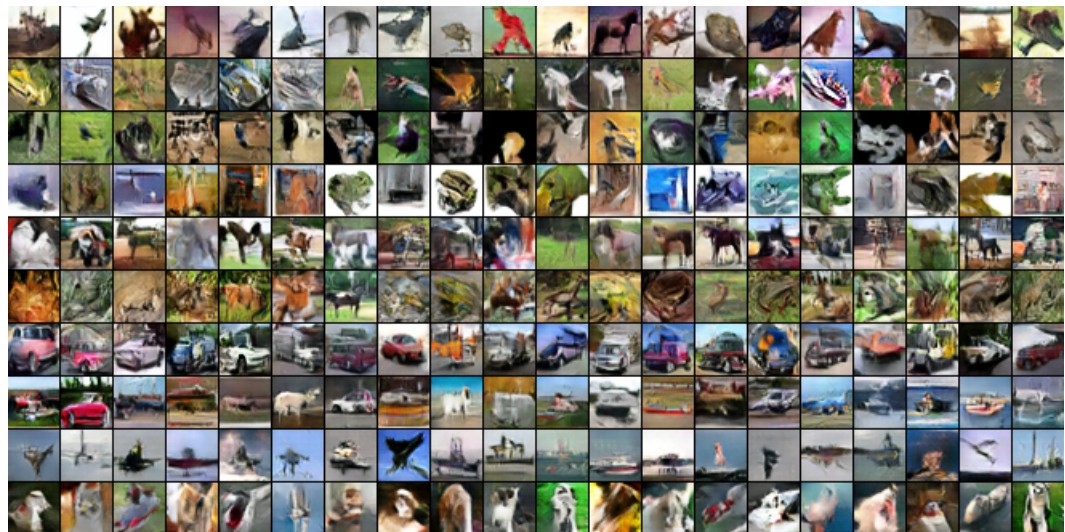

Figure 22: Images generated by **MGAN** trained on CIFAR-10 with **WGAN** objective. Images in different rows are generated by different generators.

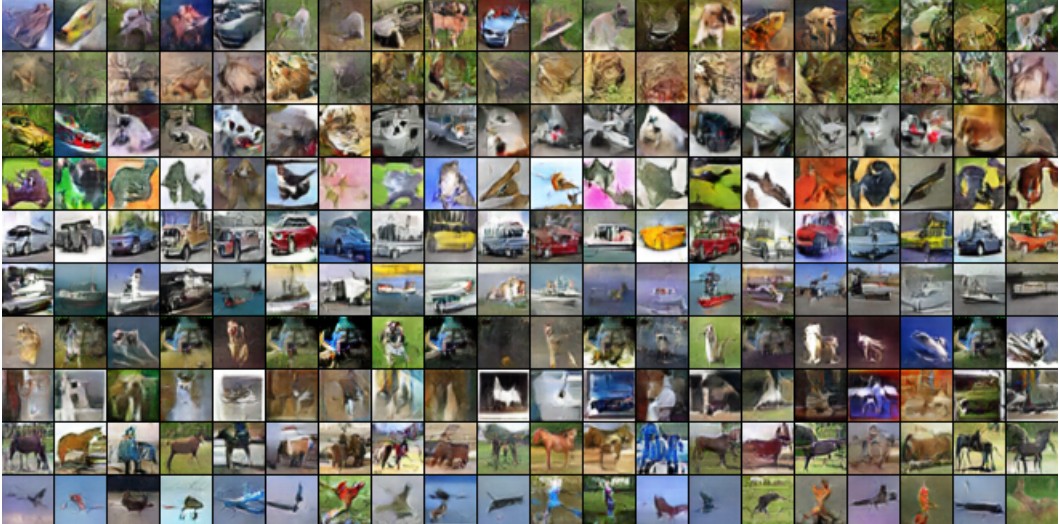

Figure 23: Images generated by **DCGAN** trained on CIFAR-10 with **GAN-NS** objective. Images in different rows are generated by different generators.

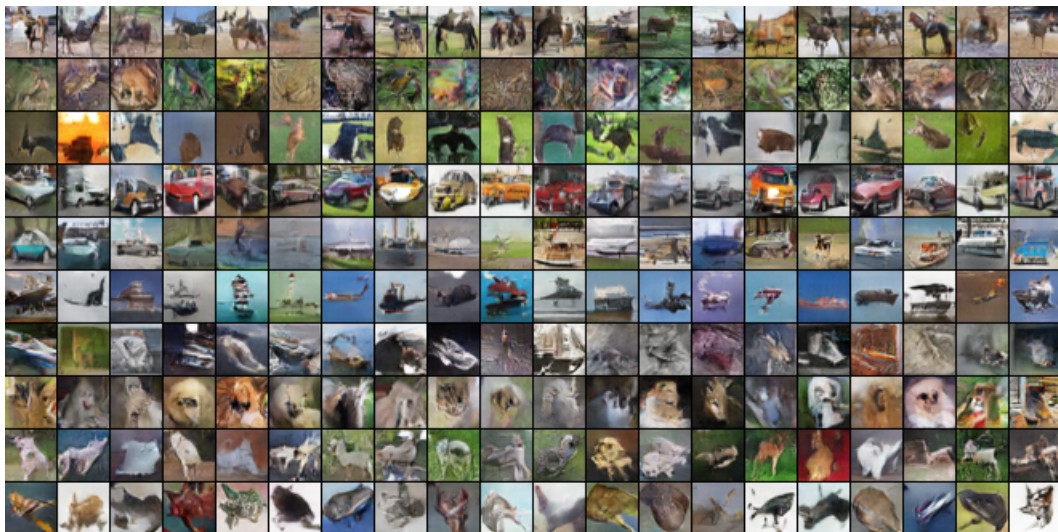

Figure 24: Images generated by **DCGAN** trained on CIFAR-10 with **LSGAN** objective.Images in different rows are generated by different generators.

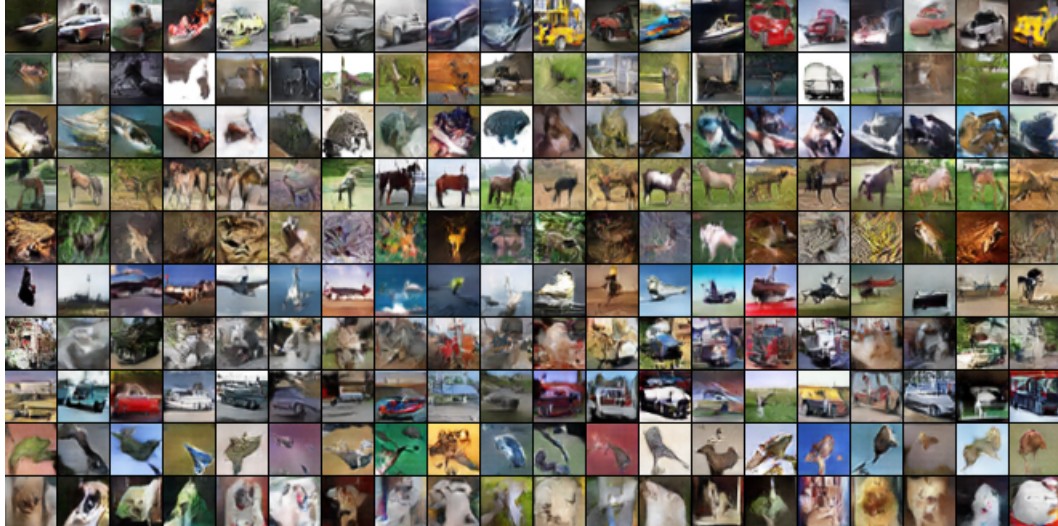

Figure 25: Images generated by **DCGAN** trained on CIFAR-10 with **GAN-MM** objective.Images in different rows are generated by different generators.

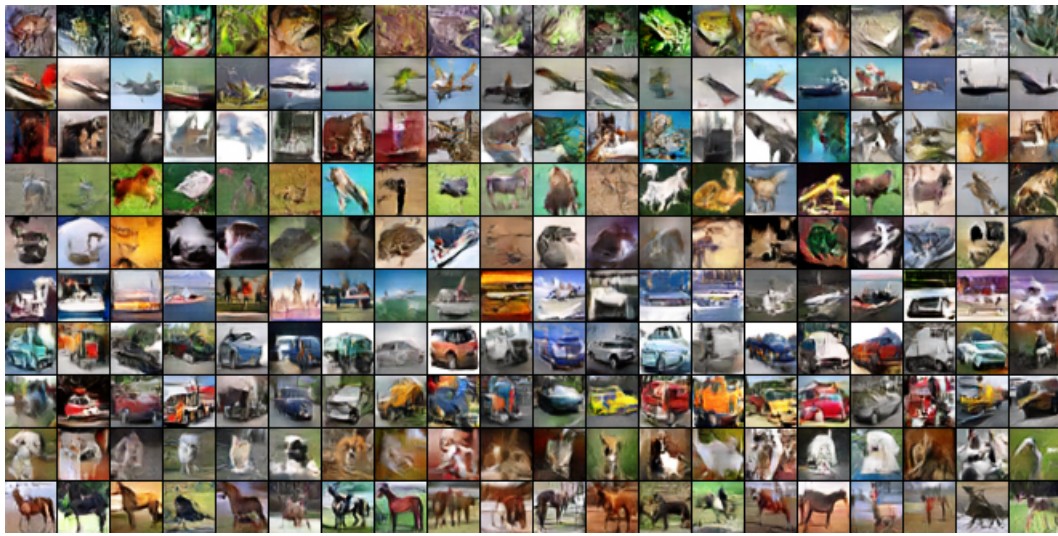

Figure 26: Images generated by **DCGAN** trained on CIFAR-10 with **WGAN** objective.Images in different rows are generated by different generators.

