# OpenReview forum: "ProbGAN: Towards Probabilistic GAN with Theoretical Guarantees"
_ICLR.cc/2019/Conference_

### Official Review · AnonReviewer1 · 2018-11-01
**A Bayesian GAN with where data distribution is an equilibrium**

**Rating:** 9
**Confidence:** 4

**Review:**

Mode collapse in the context of GANs occurs when the generator only learns one of the
multiple modes of the target distribution. Mode collapsed can be tackled, for instance, using Wasserstein distance instead of Jensen-Shannon divergence. However, this sacrifices accuracy of the generated samples.

This paper is positioned in the context of Bayesian GANs (Saatsci & Wilson 2017) which, by placing a posterior distribution over the generative and discriminative parameters, can potentially learn all the modes. In particular, the paper proposes a Bayesian GAN that, unlike previous Bayesian GANs, has theoretical guarantees of convergence to the real distribution.

The authors put likelihoods over the generator and discriminator with logarithms proportional to the traditional GAN objective functions. Then they choose a prior in the generative parameters which is the output of the last iteration. The prior over the discriminative parameters is a uniform improper prior (constant from minus to plus infinity). Under this specifications, they demonstrate that the true data distribution is an equilibrium under this scheme.

For the inference, they adapt the Stochastic Gradient HMC used by Saatsci & Wilson. To approximate the gradient of the discriminator, they take samples of the generator parameters. To approximate the gradient of the generator they take samples of the discriminator parameters but they also need to compute a gradient of the previous generator distribution. However, because this generator distribution is not available in close form they propose two simple approximations.

Overall, I enjoyed reading this paper. It is well written and easy to follow. The motivation is clear, and the contribution is significant. The experiments are convincing enough, comparing their method with Saatsci's Bayesian GAN and with the state of the art of GAN that deals with mode collapse. It seems an interesting improvement over the original Bayesian GAN with theoretical guarantees and an easy implementation.

Some typos:

- The authors argue that compare to point mass...
+ The authors argue that, compared to point mass...

- Theorem 1 states that any the ideal generator
+ Theorem 1 states that any ideal generator

- Assume the GAN objective and the discriminator space are symmetry
+ Assume the GAN objective and the discriminator space have symmetry

- Eqn. 8 will degenerated as a Gibbs sampling
+ Eqn. 8 will degenerate as a Gibbs sampling

---

> ### Author Response · Authors · 2018-11-17
> **Response to  AnonReviewer1**
>
> Dear AnonReviewer1,
>
> Thank you for agreeing with the significance of our contribution and voting to accept our paper. We will address the typos.
>
> We make an additional remark here, which might be interesting. Bayesian modeling has been introduced in several mini-max problems in the deep learning community, such as adversarial (robust) learning [1] and GANs. However, most prior works pose Bayesian method as a heuristic without theoretical analysis. This work presents an important initial step toward a rigorous study of modernized Bayesian approaches.
>
> [1] Nanyang Ye, Zhanxing Zhu. Bayesian Adversarial Learning. 32nd Annual Conference on Neural Information Processing Systems (NIPS 2018)

---

### Official Review · AnonReviewer2 · 2018-11-01
**Stripping the priors from Bayesian GANs**

**Rating:** 5
**Confidence:** 4

**Review:**

Summary
=========
The paper extends Bayesian GANs by altering the generator and discriminator parameter likelihood distributions and their respective priors.
The authors further propose an SGHMC algorithm to collect samples of the resulting posterior distributions on each parameter set and evaluate their approach on both a synthetic and the CIFAR-10 data set.
They claim superiority of their method, reporting a higher distance to mode centers of generated data points and better generator space coverage for the synthetic data set and better inception scores for the real world data for their method.

Review
=========
As an overall comment, I found the language poor, at times misleading.
The authors should have their manuscript proof-read for grammar and vocabulary.
Examples:
- amazing superiority (page 1, 3rd paragraph)
- Accutally... (page 1, end of 3rd paragraph)
- the total mixture of generated data distribution (page 3, mid of 3.1)
- Similarity we define (page 3, end of 3.1)
- etc.
Over the whole manuscript, determiners are missing.

The authors start out with a general introduction to GANs and Bayesian GANs in particular,
arguing that it is an open research question whether the generator converges to the true data generating distribution in Bayesian GANs.
I do not agree here. The Bayesian GAN defines a posterior distribution for the generator that
is proportional to the likelihood that the discriminator assigns to generated samples.
The better the generator, the higher the likelihood that the discriminator assign to these samples.
In the case of a perfect generator, here the discriminator is equally unable to distinguish real and generated samples and consequently degenerates to a constant function.
Using the same symmetry argument as the authors, one can show that this is the case for Bayesian GANs.

While defining the likelihood functions, the iterator variable t is used without introduction.

Further, I a confused by their argument of incompatibility.
First, they derive a Gibbs style update scheme based on single samples for generator and discriminator parameters using
posteriors in which the noise has been explicitly marginalized out by utilizing a Monte Carlo estimate.
Second, the used posteriors are conditional distributions with non-identical conditioning sets.
I doubt that the argument still holds under this setting.

With respect to the remaining difference between the proposed approach and Bayesian GAN,
I'd like the authors elaborate where exactly the difference between expectation of objective value
and objective value of expectation is.
Since the original GAN objectives used for crafting the likelihoods are deterministic functions,
randomness is introduced by the distributions over the generator and discriminator parameters.
I would have guessed that expectations propagate into the objective functions.

It is, however, interesting to analyze the proposed inference algorithm, especially the introduced posterior distributions.
For the discriminator, this correspond simply to the likelihood function.
For the generator, the likelihood is combined with some prior for which no closed form solution exists.
In fact, this prior changes between iterations of the inference algorithm.
The resulting gradient of the posterior decomposes into the gradient of the current objective and the sum over all previous gradients.
While this is not a prior in the Bayesian sense (i.e. in the sense of an actual prior belief), it would be interesting to have a closer look at the effect this has on the sampling method.
My educated guess is, that this conceptually adds up to the momentum term in SGHMC and thus slows down the exploration of the parameter space and results in better coverage.

The experiments are inspired by the ones done in the original Bayesian GAN publication.
I liked the developed method to measure coverage of the generator space although I find the
term of hit error misleading.
Given that the probabilistic methods all achieve a hit rate of 1, a lower hit error actually points to worse coverage.
I was surprised to see that hit error and coverage are only not consistently negatively correlated.
Adding statistics over several runs of the models (e.g. 10) would strengthen the claim of superior performance.

---

> ### Author Response · Authors · 2018-11-17
> **Response to  AnonReviewer2**
>
>
> Dear AnonReviewer2,
>
> Thank you for the feedback.
> Following is our response to your concerns.
>
> === convergence of Bayesian GAN ===
>
> The convergence of Bayesian GAN is indeed a problem, which is one of our key contributions. Bayesian GAN has a subtle difference from the original GANs during learning. To compute the posterior, Bayesian GAN cannot be learned by vanilla gradient descent methods, but is learned by SGHMC. In SGHMC framework, the gradient is always adulterated by white noises. Thus if the gradient from discriminator is always zero, the generator distribution will converge to a Gaussian distribution instead of staying unchanged.
>
> In contrast, we fix this issue by a well-crafted prior for the generator distribution. Intuitively, the gradient from the prior helps combat with the noise and prevent degeneracy of the generator distribution towards a Gaussian distribution. Please note theorem 1 does not hold without introducing suitable prior for the generator.
>
>
> === expectation of objective value v.s. objective value of expectation ===
>
> This difference is another very critical improvement from Bayesian GAN. We will make it more clear in the revision of the paper.
>
> As shown in Eqn 8, to compute likelihood, Bayesian GAN takes expectation after computing the GAN objective value. While as shown in Eqn 2, we compute GAN objective value after the expectation. The subtle adjustment is crucial. Theorem 1 will not hold if the likelihood is defined as the expectation of loss value as Bayesian GAN did. Intuitively, because the expectation \E_{q_g} p_{gen}(x;\theta_g)) is equivalent to the data distribution p_model(x) produced by the generator distribution, it makes sense to compute GAN objective over it instead of the reversed order (in Bayesian GAN). Besides, it’s easy to see the gradients of the two different likelihoods is different since, for a given function f, the gradient of \sum_i f(x_i) is usually different from that of f(\sum_i x_i).
>
> === clarification on incompatibility ===
>
> The incompatibility corresponds to the incompatibility between two conditional distributions that can not belong to the same joint distribution. We identify a theoretical flaw of Bayesian GAN under a very simple setting (when only using single Monte-Carlo sample) that leads to incompatible conditionals of generator and discriminator. Moreover, we are not very certain about the concern “the used posteriors are conditional distributions with non-identical conditioning sets. I doubt that the argument still holds under this setting.” Further explanation about “non-identical conditioning sets” will be appreciated.
>
> === relationship between hit error and coverage ===
>
> By our definition, ‘hit error’ is the averaged distance between the generated data points (projected into a low dimensional space) and the low dimensional hyperplane that the ground truth mode lies in. While the ‘coverage error’ measures the similarity between the distribution of projected data points and the ground truth data distribution which is uniform.
>
> Note that these two metrics are actually orthogonal to each other, due to the fundamental difference between projection distances (‘hit error’) and how the projections are distributed (’coverage error‘). It’s possible to get the same projection distances in a scattered or dense way. It’s also possible to get the same projections from different projection distances.
>
> We will change the terminology ‘hit error’ to ‘hit distance’ to make it clearer in our revision.
>
> === further analyze of our inference algorithm ===
>
> The momentum explanation seems an interesting direction to yield a formal explanation of such approximations, but we do not have a concrete analysis yet and leave it as future work.

---

> > ### Comment · AnonReviewer2 · 2018-11-28
> > **rating update**
> >
> > Thanks for your responses!
> >
> > I really like the way you clarify the differences in expectation over objective vs. objective of expected values. However, if you compare your method to the BGAN, you should use the priors as defined in their paper (i.e. broad normals in theirs vs. uniform in yours) when characterizing the BGAN.
> >
> > As I mentioned in my initial review, I think you factually strip all priors from the model formulation that are independent of data. In my understanding, this is the basis of any Bayesian model, a data-independent prior that encodes prior belief on the parameters that is then updated by the data via the likelihood.
> > The implicit uniform prior on your discriminator distribution might be interpreted as non-informative (one could argue it to be a Jeffreys prior) but the definition of the prior on the generator distribution being the posterior state of the last update step is rather unorthodox and, more importantly, relying on data.
> > This might strip the inherent property of robustness to overfitting from the model which should be one of the main reasons to formulate a Bayesian model in the first place and I think that more elaboration on why this is still a Bayesian model (as they claim in the paper title) is needed here.
> >
> > I totally agree with Rev3 that this is nice model with impressive results, I'm just not convinced by the explanation of why this is. As is argued in the original Bayesian GAN paper, having uniform priors is effectively the same as using a classical GAN. One could see your approach as a clever (probabilistic) extension of the optimization procedure of the classical GAN.
> >
> > Given that you have clarified on other of my concerns, I updated my rating.

---

> > > ### Author Response · Authors · 2018-12-06
> > > **Thanks for your helpful comments.**
> > >
> > > Following is our response to your updated comments.
> > >
> > > === Bayesian GAN prior in toy experiment ===
> > >
> > > We agree that using a broad normal prior would be a better choice. We have rerun our experiment with the normal prior (with mean 0, std 1) and got the similar results as in the uniform prior case. We will certainly include the new result in our next version of the paper.
> > >
> > > Following is a remark on the modification we make on the toy model to use normal prior. We reparameterize the generator and discriminator. Generator with parameter theta^g produces a data distribution p(x_i; theta^g) = exp(theta^g_i) / sum_j exp(theta^d_j). Under a normal prior N(0,1), its prior probability is p(theta^g) = \prod_j exp(- theta^g * theta^g / 2). Discriminator with parameter theta^d has a score function D(x_i; theta^d) = sigmoid(theta^d_i). Under a normal prior N(0,1), its prior probability is p(theta^d) = \prod_j exp(- theta^d * theta^d / 2).
> > >
> > > === why to strip the normal prior ===
> > >
> > > It is a very good point. Thank you for your insightful comment on it.
> > >
> > > It is true that the prior is crucial for a Bayesian model and should encode domain knowledge. What our empirical analysis shows is that a Gaussian prior is not helpful in the task of Bayesian GANs. At least, the normal prior does not show an advantage over the non-informative prior. Intuitively, putting a normal prior is very similar to have an L2 regularization when training a neural network. It looks helpful in the sense of robust to overfitting. However, we remark that, unlike typical supervised learning where model fitting is connected to generalization performance, an "overfitting" model is desirable for GANs that matches the data distribution perfectly.
> > >
> > > Since the normal prior does not work, we need a more involved prior that makes the Bayesian modeling work. Our solution is the “unorthodox” generator prior as you mentioned. Although it looks rather “unorthodox” at first blush, this generator prior is standard in the following senses: (1) As we previously explained to R3, our Bayesian model actually includes two separate models, one for the generator and one for the discriminator. Hence from the generator’s perspective, the generator is the ‘model’ and the discriminator is the ‘data’ and the other way around from the discriminator's perspective. Note that the real data distribution we want to learn is actually a third-party component; it is, therefore, proper to involve the real data in the prior. (2) Our generator prior encodes our prior belief in the sense that the generator distribution should be stable if the discriminator cannot distinguish the synthetic data and the real data well.
> > >
> > > === robustness to overfitting ===
> > >
> > > We want to emphasize that the setting we are handling with is different from the traditional prediction settings where Bayesian methods are applied commonly. When dealing with classification or regression, robustness to overfitting is quite important. However, in the GAN computation setting, the overfitting issue is not the main concern since the goal is to produce a distribution that matches the real data distribution perfectly, rather than generalizing to unseen data.

---

### Official Review · AnonReviewer3 · 2018-11-03
**experimental work now conclusive**

**Rating:** 6
**Confidence:** 3

**Review:**

PRIOR COMMENT:   This paper should be rejected based on the experimental work.
Experiments need to be reported for larger datasets.  Note the MGAN
paper reports results on STL-10 and ImageNet as well.

NOTE:  this was addressed by the 27/11 revision, which included good
   results for these other data sets, thus I now withdraw the comment

Note, your results on CIFAR-10 are quite different to those in the
MGAN paper.  Your inceptions scores are worse and FIDs are better!!  I
expect you have different configurations to their paper, but it would
be good for this to be explained.  NOTE:   explained in response!

NOTE:  this was addressed by the 27/11 revision

I thought the related work section was fabulous, and as an extension
to BGAN, the paper is a very nice idea.  So I benefited a lot from reading
the paper.

I have some comments on Bayesian treatment.  In Bayesian theory, the
true distribution $p_{data}$ cannot appear in any evaluated formulas,
as you have it there in Eqn (1) which is subsequently used in your
likelihood Eqn (2).  Likelihoods are models and cannot involve "truth".

Lemma 1:  Very nice observation!!  I was trying to work that out,
once I got to Eqn (3), and you thought of it.

Also, you do need to explain 3.2 better.  The BGAN paper, actually, is
a bit confusing from a strict Bayesian perspective, though for
different reasons.  The problem you are looking at is not a
time-series problem, so it is a bit confusing to be defining it as
such.  You talk about an iterative Bayesian model with priors and
likelihoods.  Well, maybe that can be *defined* as a probabilistic
model, but it is not in any sense a Bayesian model for the estimation
of $p_{model}$.

NOTE:  anonreviewer2 expands more on this

What you do with Equation (3) is define a distribution on
$q_g(\theta_g)$ and $q_d(\theta_d)$ (which, confusingly, involves the
"true" data distribution ... impossible for a Bayesian formulation).
You are doing a natural extension of the BGAN papers formulation in
their Eqs (1) and (2).  This, as is alluded to in Lemma 1.  Your
formulation is in terms of two conditional distributions, so
conditions should be given that their is an underlying joint
distribution that agrees with these.  Lemma 1 gives a negative result.
You have defined it as a time series problem, and apparantly one wants
this to converge, as in Gibbs sampling style.  Like BGAN, you have
just arbitrarily defined a "likelihood".

To me, this isn't a Bayesian model of the unsupervised learning task,
its a probabilistic style optimisation for it, in the sense that you are defining a probability
distribution (over $q_g(\theta_g)$ and $q_d(\theta_d)$) and sampling
from it, but its not really a "likelihood" in the formal sense.  A
likelhood defines how data is generated.  Your "likelihood" is over
model parameters, and you seem to have ignored the data likelihood,
which you define in sct 3.1 as $p_{model}()$.

Anyway, I'm happy to go with this sort of formulation, but I think you
need to call it what it is, and it is not Bayesian in the standard sense.  The theoretical
treatment needs a lot of cleaning up.  What you have defined is a
probabilistic time-series on $q_g(\theta_g)$ and $q_d(\theta_d)$.
Fair enough, thats OK.  But you need to show that it actually works in
the estimation of $p_{model}$.  Because one never has $p_{data}$, all
your Theorem 1 does is show that asymptotically, your method works.
Unfortunately, I can say the same for many crude algorithms, and most
of the existing published work.  Thus, we're left with requiring a
substantial empirical validation to demonstrate the method is useful.

Now my apologies to you: I could make somewhat related statements
about the theory of the BGAN paper, and they got to publish theirs at
ICLR!  But they did do more experimentation.

Oh, and some smaller but noticable grammar/word usage issues.

NOTE:  thanks for your good explanation of the Bayesian aspects of the model ...
yes I agree, you have a good Bayesian model of the GAN computation , but it
is still not a Bayesian model of the unsupervised inference task.  This is a somewhat
minor point, and should not in anyway influence worth of the paper ... but clarification
in paper would be nice.

---

> ### Author Response · Authors · 2018-11-18
> **Response to  AnonReviewer3**
>
> Dear AnonReviewer3,
>
> Thank you for the insightful comments.
> Following is our response to your concerns.
>
> === experiments ===
>
> We will include results on STL-10 and ImageNet in the revision, or a later version if our machines cannot catch up the rebuttal deadline. Compared with Bayesian GAN, actually, we did a more thorough study on the choice of objective function, and our synthetic dataset is harder and more illustrative.
>
> Here we clarify the discrepancy between our quantitative evaluation of MGAN and that of the original paper. We actually use the official open-sourced code of MGAN with the same configurations (model architectures, training data). The discrepancy comes from the inception model used to compute FID. We compute FID with PyTorch Inception model (https://github.com/mseitzer/pytorch-fid.). The original MGAN paper did not say which inception model they have used. Our guess is that they used the Tensorflow inception model (https://github.com/bioinf-jku/TTUR). We observed FID computed by PyTorch model is much lower than that computed by the Tensorflow model, because of the different weights of the pre-trained models. A similar phenomenon has been recently observed for Inception Score [1]. To favor a more complete comparison, we will update our FID results by switching to the Tensorflow version.
>
> We had posted the updated results in the comment. In our experiments, the MGAN with GAN-NS objective has the same setting with original MGAN. The Inception score and FID we get are 7.25 and 27.55 which are both worse than the scores reported in the original paper, 8.33 and 26.7. We train MGAN with the officially released code under the configuration reported in the MGAN paper (Table 4 in the appendix). The scores we reported is the best we can get via several training trials.
>
> [1] Barratt, Shane, and Rishi Sharma. "A Note on the Inception Score." arXiv preprint arXiv:1801.01973 (2018).
>
> === Bayesian formulation ===
>
> Our method has two separate Bayesian models, one for the generator and one for the discriminator. Take the Bayesian perspective for the generator as an example. The likelihood defined in the first equation of Eqn 2 gives the probability of observing some fixed discriminator distribution for some generator parameter, i.e., p(D^{(t)} | \theta_g). Composite with the prior of the generator parameter q^{(t)}(\theta_g), it is a Bayesian model from a strict perspective. Indeed, to see the correspondence of ‘model parameter’ and ‘data’  in classic Bayesian theory, our generator is the ‘model’ and the discriminator is the ‘data’. We estimate generator distribution by the observed discriminator distribution.
>
> The novelty from classic Bayesian models is on the inference procedure. We integrate the two standard Bayesian models into a dynamical system: each Bayesian problem is solved alternatingly. From a game-theoretic point of view, each optimization problem is the best response strategy of the corresponding player, and the equilibrium presents a generator distribution that produces the target data distribution.
>
> === Why time-series modelling ===
>
> The problem is not a time-series problem. We simply solve it in an iterative manner. (akin to SGD that can iteratively solve both time-series and non-time-series problems). Our goal is to find the equilibrium of generator and discriminator distributions, where they satisfy each other’s posterior under our Bayesian criterion. It is, however, possible to find the equilibrium via an iterative scheme. We will make this part more clear in the revision.
>
> === A clarification about theorem 1 ===
>
> It is indeed true that Theorem 1 only shows an analysis of the optimal solution in an asymptotic scenario. Unfortunately, it is, to our best knowledge, the best property that has been obtained in recent literature on GANs [2, 3, 4, 5, 6]. However, please note that Bayesian GAN does not even possess such asymptotic property and the difficulty of avoiding such problem as revealed by our analysis in Section 4.2. In contrast, our method is to the first Bayesian method to establish such property.
>
> [2] Goodfellow, Ian, et al. "Generative adversarial nets." Advances in neural information processing systems. (NIPS 2014)
> [3] Hoang, Quan, et al. "MGAN: Training generative adversarial nets with multiple generators." (ICLR 2018)
> [4] Arjovsky, Martin, Soumith Chintala, and Léon Bottou. "Wasserstein generative adversarial networks."(ICML 2017)
> [5] Mao, Xudong, et al. "Least squares generative adversarial networks." Computer Vision (ICCV), 2017 IEEE International Conference on. IEEE, 2017.
> [6] Zhao, Junbo, Michael Mathieu, and Yann LeCun. "Energy-based generative adversarial network." (ICLR 2017)

---

> ### Author Response · Authors · 2018-11-27
> **Thanks again for your thoughtful comments.**
>
> #1 You have a good Bayesian model of the GAN computation, but it is still not a Bayesian model of the unsupervised inference task.
>
> Yes, you are right. In this work, we aim to develop a better Bayesian model of the GAN computation. Generally, Bayesian models for unsupervised inference tasks could be a larger topic.
>
> #2 I want to see results on the big data sets.
>
> Thanks for being positive of our work. We have included the results on STL-10 and ImageNet in our revision of the paper (e.g., Table 4 and Figure 4 of Section 5.2). As mentioned in the general response above, our model does provide better performance on both datasets with significant improvements of FID scores. We hope that with the additional results, the experimental work in the current version is more conclusive.

---

### Author Response · Authors · 2018-11-18
**Updating Inception and Frechet Inception Distance Results on CIFAR10. (Table 3 in the paper)**

Previously, our FID results are computed using a PyTorch Implementation (https://github.com/mseitzer/pytorch-fid). Note that there exists a large discrepancy between the FID results conducted by PyTorch Inception model and Tensorflow model. Hence, to facilitate the comparison with previous paper, we decide to reevaluate by the official Tensorflow FID computation code (https://github.com/bioinf-jku/TTUR).

Here are the updated results.

                       Inception scores (higher is better)
                  GAN-MM  & GAN-NS & WGAN & LSGAN
DCGAN    &   6.53     &        7.21 &      7.19 &      7.36
MGAN      &   7.19     &        7.25 &      7.18 &      7.34
BGAN       &   7.21    &        7.37 &       7.26 &      7.46
ours-PSA  &  7.75     &       7.53  &      7.28 &      7.36

                             FIDs (lower is better)
                  GAN-MM  & GAN-NS & WGAN & LSGAN
DCGAN    &      35.57 &     27.68 &    28.31 &    29.11
MGAN      &      30.01 &     27.55 &    28.37 &    30.72
BGAN       &      29.87 &     24.32 &    29.87 &    29.19
ours-PSA  &     24.60  &    23.55 &     27.46 &    26.90

Note that we are reporting the results with the highest ‘Inception score - 0.1 FID’ for each model. Thus the Inception scores results are also updated.

---

### Author Response · Authors · 2018-11-27
**General Response**

We thank all the reviewers for the insightful comments and helpful suggestions. Here we summarize the major changes we did in the revision of our paper.

1.  Adding experiment results on STL-10 and ImageNet.

We follow R3’s suggestion to compare our models and baselines (MGAN, BGAN) on the larger datasets. Our model does provide better performance on both datasets. Especially, the improvement of FID scores looks significant. We include the new experiment results (Table 4 and Figure 4) in Section 5.2.

2. Updating Inception score and FID results on CIFAR-10.

Thanks to R3’s help, we find the discrepancy between FIDs given by the PyTorch model and the Tensorflow model. We have switched to the official Tensorflow model for evaluation and updated all results in Table 3 (Section 5.2). We also put a remark in Section B.1 (of the appendix) to make it clearer.

3. Emphasizing the difference between our model and Bayesian GAN.

R2 suggests that we elaborate more about the difference between our likelihood design (objective value of expectation) and Bayesian GAN’s likelihood (expectation of objective value). We revise Section 4.2 to explain the differences both in the likelihood and in the prior more clearly.

4. Adding a toy experiment to demonstrate different convergence behavior of our model and Bayesian GAN (Figure 1).

We include a new toy experiment on categorical distributions as empirical support for the superior convergence property of our model over the Bayesian GAN.

In our toy experiment,  the data is sampled from a finite discrete space (more specifically, a categorical distribution). It is ideal to examine the Bayesian formulation in a finite case since the posterior can then be computed analytically and does not have error caused by inference algorithms. We try different combinations of likelihoods and priors in the experiment and compare their learned distributions.

In Figure 1, we visualize the generated data distributions of different models after they converge. The results show that only when using the combination of our likelihood and our prior can the model converge to the correct equilibrium. The full details of the experiment are included in Section D (of the appendix). This example also serves as an illustration of the convergence issue of Bayesian GAN.

Minor changes:

1. Change the term ‘hit error’ to ‘hit distance’ (e.g., in Table 2) to avoid the potential misunderstanding of its meaning.

2. Add a few sentences in Section 4.1 to explain why Theorem 1 does not hold for Bayesian GAN.

---

### Meta-Review · Area_Chair1 · 2018-12-14
**Interesting new model with good performance**

**Confidence:** 4
**Recommendation:** Accept (Poster)

**Metareview:**

The paper proposes a new method that builds on the Bayesian modelling framework for GANs and is supported by a theoretical analysis and an empirical evaluation that shows very promising results. All reviewers agree, that the method is interesting and the results are convincing, but that the model does not really fit in the standard Bayesian setting due to a data dependency of the priors. I would therefore encourage the authors to reflect this by adapting the title and making the differences more clear in the camera ready version.